# Flow for Future: Geometric SE(3)-Equivariant Flow Matching for 3D Trajectory Prediction

**Junwei Wu** [1] **Yihang Liu** [2] **Ruixuan Yu** [1] **Jian Sun** [3 4]

## Abstract

Predicting 3D geometric trajectory requires capturing complex spatiotemporal dependencies while preserving physical symmetries. While flow matching offers a powerful generative paradigm, extending it to SE(3)-equivariant dynamics is challenging due to the inherent gap between deterministic history and stochastic evolving flows. To address this, we introduce **GSE-Flow**, an SE(3)-equivariant flow matching framework. We first propose a *Coherent Sequence Encoding* and *Time-Modulated Embedding* strategy that unifies historical and evolving streams, incorporating velocity and flow time via equivariant affine transformations to guide continuous evolution. We further design a *Geometry-Feature Tensorization* mechanism that projects node states into a tensor product space, enabling *Context-Flow Fusion* to guide trajectory evolution with historical context. GSE-Flow guarantees theoretical SE(3)-equivariance and achieves SOTA accuracy on MD17, MD22, and CMU MoCap benchmarks for geometric trajectory prediction, while demonstrating generality by enhancing deterministic baselines. Code is available at `https://github.com/aegine/GSE-Flow`.

## 1. Introduction

SE(3)-equivariant trajectory prediction aims to forecast the future motions of geometric systems given historical observations. This task is widely applied in fields ranging from molecular simulation (Huang et al., 2025; Shrestha & Fu, 2025) to autonomous driving (Peri et al., 2022; Madjid et al., 2025). A critical requirement is SE(3)-equivariance, which means that the predicted future trajectories undergo the same translation or rotation applied to historical observations.

Significant strides have been made in embedding geometric symmetry into motion forecasting. Early foundational works established distinct paradigms. One stream of research relies on computationally intensive higher-order irreducible representations, as seen in TFN (Thomas et al., 2018) and SE(3)-TR (Fuchs et al., 2020). Another approach utilizes efficient invariant scalarization or vector neurons, exemplified by methods like EGNN (Satorras et al., 2021), GMN (Huang et al., 2022), EGHN (Han et al., 2022), and VN (Deng et al., 2021). More recently, a third paradigm based on pose canonicalization and averaging has emerged, with representative works including FAGNN (Puny et al., 2022), PT-EvNet (Yu & Sun, 2024), and FMN (Huang et al., 2025). To extend these spatial architectures to dynamic systems, research has evolved toward sophisticated temporal kinematic modeling (Xu et al., 2023; Wu et al., 2023; Xu et al., 2024). Despite these innovations, the dominant paradigm remains rooted in deterministic regression. They typically treat geometric trajectory prediction as mapping from observations to future states, often overlooking the continuous temporal evolution inherent in dynamic systems.

To bridge the gap between deterministic regression and continuous physical dynamics, Flow Matching (FM) offers a principled solution by learning instantaneous vector fields, naturally recovering temporal dynamics (Chen et al., 2018; Lipman et al., 2023; Liu et al., 2023; Albergo et al., 2025). Recent advancements have successfully applied FM to general motion estimation, effectively capturing the stochastic multimodality of human and vehicle trajectories (Hu et al., 2023; Fu et al., 2025; Yan et al., 2025). However, these frameworks predominantly operate in absolute coordinate systems, and often struggle to generalize across rotated reference frames. Conversely, while equivariant FM has significantly advanced static molecular generation (Köhler et al., 2020; Klein et al., 2023; Song et al., 2023; Hassan et al., 2024), extending it to history-conditioned trajectory prediction is non-trivial. Unlike generating static structures from invariant scalars, trajectory forecasting requires conditioning on dynamic geometric sequences to maintain kinematic

---

[1]School of Airspace Science and Engineering, Shandong University, Weihai, 264209, China [2]School of Mathematics and Statistics, Shandong University, Weihai, 264209, China [3]School of Mathematics and Statistics, Xi'an Jiaotong University, Xi'an, 710049, China [4]Pazhou Laboratory (Huangpu), Guangzhou, Guangdong, 510555, China. Correspondence to: Ruixuan Yu <yuruixuan@sdu.edu.cn>.

*Proceedings of the 43rd International Conference on Machine Learning*, Seoul, South Korea. PMLR 306, 2026. Copyright 2026 by the author(s).

continuity with the observed past.

To address these limitations, we propose GSE-Flow, a geometric SE(3)-equivariant flow matching framework for 3D trajectory prediction. By parameterizing the instantaneous flow state via coherent sequence encoding, time-modulated embeddings, and context-flow fusion, GSE-Flow models trajectory evolution while strictly enforcing symmetry. This work represents a novel framework in leveraging flow matching for precise and stable 3D geometric trajectory prediction. The contributions are summarized as follows:

- **Generative Equivariant Framework:** We propose **GSE-Flow**, a novel framework that integrates conditional flow matching with SE(3)-equivariant representation learning. By formulating trajectory forecasting as a continuous generative flow process, it ensures strict physical symmetry and modeling effectiveness, elevating deterministic regression baselines to a generalizable generative paradigm.

- **Consistent Flow Backbone:** We develop a specialized equivariant backbone that synergizes temporal evolution with contextual interaction. It utilizes Time-Modulated Embeddings and Coherent Sequence Encoding to fuse evolution time with sequential features, while employing Geometry-Feature Tensorization and Context-Flow Attention to bridge the evolving flow state with historical context.

- **State-of-the-Art Performance:** Extensive experiments on standard benchmarks across varying scales and scenarios demonstrate that GSE-Flow significantly outperforms existing baselines for SE(3)-equivariant geometric trajectory prediction. Furthermore, it exhibits strong generality, enhancing deterministic backbones to achieve consistent performance gains.

## 2. Related Work

**Equivariant Neural Networks.** Symmetry is a fundamental inductive bias in physical systems, necessitating models that maintain equivariance under SE(3) transformations. Early foundational research focused on high-order representations to achieve strict equivariance. Prominent works such as Tensor Field Networks (TFN) (Thomas et al., 2018), SE(3)-Transformers (Fuchs et al., 2020), and Spherical CNNs (Esteves et al., 2018) utilize spherical harmonics to model interactions between geometric tensors, while LieConv (Finzi et al., 2020) and LieTransformer (Hutchinson et al., 2021) parameterize continuous symmetries using Lie group representations. Although theoretically rigorous, these approaches rely on expensive tensor products, leading to high computational costs (Yu & Sun, 2024). To address this, efficient architectures like Vector Neurons (VN) (Deng

et al., 2021) and EGNN (Satorras et al., 2021) were introduced, operating on vector features and invariant scalar operation, respectively. Subsequent research has sought to further enhance expressiveness, where GMN (Huang et al., 2022) and EGHN (Han et al., 2022) introduce geometrical constraints and hierarchical architectures. Furthermore, FAGNN (Puny et al., 2022) utilizes frame averaging to normalize positions, while PT-EvNet (Yu & Sun, 2024) and FMN (Huang et al., 2025) explicitly learn frames to enhance feature representations. Furthermore, trajectory-specific models such as EqMotion (Xu et al., 2023), ESTAG (Wu et al., 2023), EGNO (Xu et al., 2024) and Equi-Euler GraphNet (Sharma et al., 2025) integrate temporal dynamics to capture time-dependent geometric evolutions. These deterministic architectures serve as foundational backbones for diverse orientation-sensitive tasks, enabling breakthroughs in LiDAR localization (Yang et al., 2025), aerial object detection (Wu et al., 2025), and molecular analysis (Song et al., 2023; Hassan et al., 2024; Tian et al., 2024; Eijkelboom et al., 2025). However, these deterministic methods fail to account for the inherent indeterminacy of future motion, limiting their ability to model complex trajectory distributions as emphasized in recent studies (Fu et al., 2025; Yan et al., 2025). In this paper, we propose a generative framework to capture these underlying distributions, thereby enhancing the accuracy of equivariant trajectory prediction.

**Diffusion and Flow-based Trajectory Prediction.** Recently, diffusion probabilistic model (Ho et al., 2020) and flow matching (Lipman et al., 2023) have demonstrated remarkable capabilities in trajectory prediction tasks. Specifically, diffusion-based methods like MID (Gu et al., 2022) and DifTraj (Liu et al., 2024) capture latent indeterminacy through iterative denoising processes, while BCDiff (Li et al., 2023) and SingularTrajectory (Bae et al., 2024) further enforce temporal and modal coherence via bidirectional constraints or unified representation spaces. More recently, GeoTDM (Han et al., 2024) has extended this paradigm to model geometric trajectories. To overcome the stochastic inefficiency of diffusion, flow matching has emerged as a powerful paradigm by learning the velocity field to construct approximately straight probability paths. Addressing the non-Euclidean nature of motion, RFMP (Braun et al., 2024) and MMFP (Lee et al., 2025) extend the flow prior to manifolds, ensuring predictions adhere to physical kinematic constraints, while STFlow (Brinke et al., 2025) focuses on simulating geometric dynamics.. For complex environmental interactions, TrajFlow (Yan et al., 2025), UniEgoMotion (Patel et al., 2025) and MADiff (Ma et al., 2025) condition the vector field on multi-modal contexts. For optimizing efficiency, MoFlow (Fu et al., 2025) leverages sample-driven distillation for one-step generation, while T-CFM (Ye & Gombolay, 2024) circumvents iterative denoising by learning a time-varying vector field for efficient

generation. However, existing generative methods typically fuse historical features with flow states via shallow concatenation, neglecting the deep geometric coupling essential for SE(3)-equivariant dynamics. To address this, we propose a framework that explicitly models these structural dependencies for precise generation.

## 3. Proposed Method

### 3.1. Problem Formulation & Generative Framework

**Problem Formulation.** Consider a 3D geometric system defined by historical node positions $\mathbf{P} \in \mathbb{R}^{N \times T_h \times 3}$, connectivity $\mathbf{E} \in \mathbb{R}^{N \times N}$, and SE(3)-invariant attributes $\mathbf{H} \in \mathbb{R}^{N \times T_h \times C}$ (e.g., atom types), where $N$, $T_h$, and $C$ denote the number of nodes, history frame length, and attribute dimension, respectively. Our goal is to predict future trajectory $\mathbf{F} \in \mathbb{R}^{N \times T_f \times 3}$ over a horizon $T_f$, ensuring the prediction is SE(3)-equivariant with respect to $\mathbf{P}$.

To ensure translation equivariance, we reformulate the task as predicting the relative trajectory $\mathbf{Y}$ given the spatiotemporally centered history $\mathbf{X} \equiv \mathbf{P} - \overline{\mathbf{P}}$, where the absolute future trajectory is then derived as $\mathbf{F} = \mathbf{Y} + \overline{\mathbf{P}}$.

**Trajectory Prediction via Flow Matching.** To capture the complex spatiotemporal distribution of relative trajectories, we adopt the *conditional flow matching* paradigm. The probability path that linearly interpolates between a Gaussian noise prior $\mathbf{Y}^0 \sim \mathcal{N}(\mathbf{0}, \mathbf{I})$ and the target relative trajectory $\mathbf{Y}^1 \equiv \mathbf{Y}$ is defined as:

$$\mathbf{Y}^\tau = (1-\tau)\mathbf{Y}^0 + \tau\mathbf{Y}^1, \quad \tau \in [0,1], \quad (1)$$

where $\tau$ denotes the flow time.

Unlike standard approaches that regress the instantaneous velocity field, we employ a *data-prediction* strategy for enhanced geometric stability. We parameterize our equivariant network $\mathbf{\Phi}$ to directly predict the clean target $\mathbf{Y}^1$ from the noisy intermediate state $\mathbf{Y}^\tau$. The network is optimized by minimizing the reconstruction error, which implicitly aligns the generated flow with the optimal transport path:

$$\mathcal{L} = \mathbb{E}_{\tau, \mathbf{Y}^0, \mathbf{Y}^1} \left[ ||\mathbf{Y}^1 - \mathbf{\Phi}(\mathbf{Y}^\tau, \mathbf{X}, \mathbf{H}, \mathbf{E}, \tau)||_2^2 \right]. \quad (2)$$

During inference, the trajectory is synthesized by numerically integrating the estimated vector field. Specifically, we derive the instantaneous velocity $\mathbf{u}^\tau$ from predicted target:

$$\mathbf{u}^\tau = \frac{\mathbf{\Phi}(\mathbf{Y}^\tau, \mathbf{X}, \mathbf{H}, \mathbf{E}, \tau) - \mathbf{Y}^\tau}{1 - \tau}, \quad (3)$$

and then iteratively update the state from $\mathbf{Y}^0 \sim \mathcal{N}(\mathbf{0}, \mathbf{I})$ via the Euler method:

$$\mathbf{Y}^{\tau + \Delta\tau} = \mathbf{Y}^\tau + \Delta\tau\mathbf{u}^\tau, \quad (4)$$

where $\Delta\tau$ is the step size. The final state at $\tau = 1$ serves as the predicted relative trajectory.

**GSE-Flow Framework.** Extending flow matching to equivariant 3D dynamics presents a fundamental challenge: *how to deeply couple scalar flow time and long-term historical context with evolving flow states without compromising geometric symmetry*. To address this, we propose **GSE-Flow**, a framework that instantiates the equivariant network $\mathbf{\Phi}$ to bridge the gap between historical observation and generative flow. Specifically, we first employ the **History Context Encoder (HCE)** (§ 3.2) to extract spatiotemporal geometric embeddings and SE(3)-invariant node features from observations. These outputs condition the **Flow State Encoder (FSE)** (§ 3.3), which integrates the noisy flow state with the historical sequence and injects flow time via affine modulation. Finally, the **Trajectory Generation (TG)** (§ 3.4) predicts the target state by fusing these representations, facilitating deep interaction between historical context and evolving dynamics for precise generation.

### 3.2. History Context Encoder (HCE)

The HCE module extracts high-level motion patterns from historical observations, serving as the spatiotemporal condition for generative process. To capture complex non-Markovian dependencies, we employ a backbone $\varphi$ adapted from the work of ESTAG (Wu et al., 2023) as HCE, leveraging specialized equivariant blocks (EDFT, ESM, ETM) to model frequency, spatial, and temporal dynamics. Departing from the original sequence-to-one design, we reconfigure $\varphi$ to operate in a sequence-to-sequence manner, preserving the full temporal resolution. This yields the equivariant geometric representation $\mathbf{C}_X \in \mathbb{R}^{N \times T_h \times 3}$ and $D$-dimensional invariant feature $\mathbf{C}_H \in \mathbb{R}^{N \times T_h \times D}$:

$$\mathbf{C}_X, \mathbf{C}_H = \varphi(\mathbf{X}, \mathbf{H}, \mathbf{E}). \quad (5)$$

Architectural details of $\varphi$ are provided in the Appendix D.

### 3.3. Flow State Encoder (FSE)

The FSE module is proposed to embed the noisy flow state $\mathbf{Y}^\tau$ and flow time $\tau$ into a high-level representation while preserving compatibility with the historical context. Direct integration is challenging, as naive scalar time concatenation violates the symmetry of flow states. Furthermore, the flow is inherently temporally disjoint from the history. We resolve this via two key mechanisms: *Coherent Sequence Encoding* and *Time-Modulated Embedding*.

**Coherent Sequence Encoding.** Given the flow geometric representation $\mathbf{Y}^\tau \in \mathbb{R}^{N \times T_f \times 3}$, we initialize the node features $\mathbf{Z}^\tau \in \mathbb{R}^{N \times T_f \times D}$ by replicating the last frame of the historical context $\mathbf{C}_H$ across the prediction horizon. To enforce dynamical consistency with historical observation, we first construct a unified spatiotemporal sequence by con-

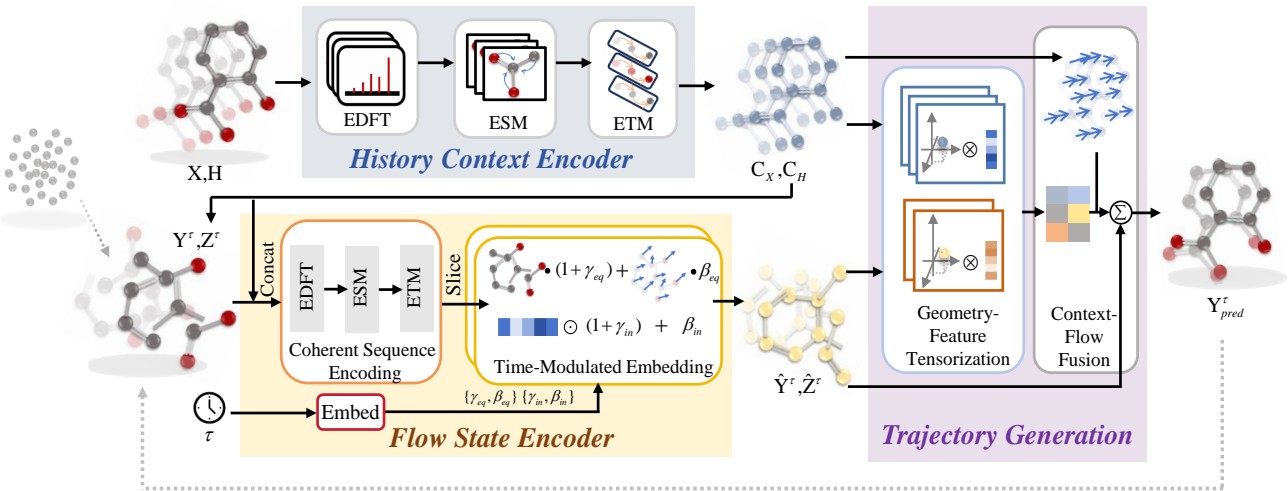

*Figure 1.* Pipeline of GSE-Flow. The framework is built on three equivariant modules: (1) History Context Encoder (HCE) for historical feature extraction; (2) Flow State Encoder (FSE) utilizing coherent sequence encoding and time-modulated embedding to capture flow dynamics; (3) Trajectory Generation (TG) that fuses history and flow features to predict the equivariant trajectory.

catenating the historical context $\{\mathbf{X}, \mathbf{C}_H\}$ with the noisy flow states $\{\mathbf{Y}^\tau, \mathbf{Z}^\tau\}$ along the temporal dimension. This extended representation, along with the connectivity $\mathbf{E}$, is then processed by the backbone $\psi$:

$$\tilde{\mathbf{Y}}^\tau, \tilde{\mathbf{Z}}^\tau = \mathrm{Slice}\big(\psi([\mathbf{X}, \mathbf{Y}^\tau], [\mathbf{C}_H, \mathbf{Z}^\tau], \mathbf{E})\big), \quad (6)$$

where $[\cdot, \cdot]$ denotes concatenation. The backbone $\psi$ adopts an equivariant architecture similar to the HCE but operates on the extended sequence. Finally, the $\mathrm{Slice}(\cdot)$ operation extracts the updated future segments $\tilde{\mathbf{Y}}^\tau \in \mathbb{R}^{N \times T_f \times 3}$ and $\tilde{\mathbf{Z}}^\tau \in \mathbb{R}^{N \times T_f \times D}$ for subsequent processing. This design facilitates consistent temporal information propagation, allowing the flow states to implicitly aggregate dynamic cues from the historical context, thereby maintaining both geometric plausibility and kinematic coherence.

**Time-Modulated Embedding.** To guide the trajectory generation, we explicitly encode the flow time $\tau$ into both geometric and semantic states of the noisy flow. For the *geometric modulation*, we first approximate the instantaneous node velocity $\tilde{\mathbf{V}}^\tau$ from the sequence $\tilde{\mathbf{Y}}^\tau$ via temporal finite differences. To incorporate kinematic trends, a time-parameterized mixing of position and velocity is proposed:

$$\hat{\mathbf{Y}}^\tau = (1 + \gamma_{eq})\tilde{\mathbf{Y}}^\tau + \beta_{eq}\tilde{\mathbf{V}}^\tau, \quad (7)$$

where $\gamma_{eq}, \beta_{eq} \in \mathbb{R}$ are time-dependent scalars computed from the embedding of $\tau$ via a Multi-Layer Perceptron (MLP). This design allows the model to dynamically weight the importance of zeroth-order (position) and first-order (velocity) cues, effectively injecting momentum-informed structural cues to guide the trajectory generation. The equivariance of $\hat{\mathbf{Y}}^\tau$ is guaranteed by following lemma.

**Lemma 3.1.** *The time-modulated position $\hat{\mathbf{Y}}^\tau$ in Eq. (7) is rotation-equivariant with respect to input position $\mathbf{X}$.*

For *feature modulation*, we inject temporal dynamics into the node features via a channel-wise affine transformation, adopting a mechanism akin to FiLM (Perez et al., 2018):

$$\hat{\mathbf{Z}}^\tau = (1 + \gamma_{in}) \odot \tilde{\mathbf{Z}}^\tau + \beta_{in}, \quad (8)$$

where $\gamma_{in}, \beta_{in} \in \mathbb{R}^D$ are time-dependent modulation vectors derived from the time embedding of $\tau$ by MLP. This operation infuses temporal evolution into the static semantic context while preserving rotation-invariance.

In summary, the FSE module establishes temporal coherence through unified sequence encoding and adapts the flow state to the flow time via affine modulation. The resulting representations $\hat{\mathbf{Y}}^\tau$ and $\hat{\mathbf{Z}}^\tau$ strictly preserve rotation-equivariance and invariance, providing time-dependent geometric and feature inputs for the deep context-flow fusion in the subsequent trajectory generation module.

### 3.4. Trajectory Generation(TG) by Context-Flow Fusion

The TG module is proposed as equivariant decoder to synthesize the final geometric trajectory by integrating the historical context from HCE with the evolved flow state from FSE. High-fidelity synthesis requires capturing long-term geometric dependencies while preserving semantic correlations. To achieve this within a rigorous equivariant framework, we devise a context-flow fusion strategy comprising three key operations: *Geometry-Feature Tensorization* to explicitly couple node geometric and semantic features, *Invariant Context-Flow Attention* to distill rotation-invariant affinity weights, and *Equivariant Trajectory Integration* to generate the future sequence via relative displacement aggregation.

**Geometry-Feature Tensorization.** To endow the rotation-invariant semantic features with explicit spatial directional-

ity, we lift the node states into a tensorial product space via the outer product of equivariant coordinates and invariant features. Formally, given the historical context $\{\mathbf{C}_X, \mathbf{C}_H\}$ and the future flow state $\{\hat{\mathbf{Y}}^\tau, \hat{\mathbf{Z}}^\tau\}$, we construct the geometry-feature tensors $\mathbf{Q}_{\text{hist}} \in \mathbb{R}^{N \times T_h \times 3D}$ and $\mathbf{Q}_{\text{flow}}^\tau \in \mathbb{R}^{N \times T_f \times 3D}$ with the elements as:

$$\begin{aligned}
\mathbf{Q}_{\text{hist}}(n, t_h) &= \text{vec}\big(\mathbf{C}_X(n, t_h) \otimes \mathbf{C}_H(n, t_h)\big), \\
\mathbf{Q}_{\text{flow}}^\tau(n, t_f) &= \text{vec}\big(\hat{\mathbf{Y}}^\tau(n, t_f) \otimes \hat{\mathbf{Z}}^\tau(n, t_f)\big),
\end{aligned} \tag{9}$$

where $\otimes$ denotes the outer product, $\text{vec}(\cdot)$ is the vectorization operation, and indices $n, t_h, t_f$ denote the node, history frame and future frames, respectively. This formulation yields a rotation-covariant descriptor, where spatial orientation explicitly modulates semantic correlations.

**Invariant Context-Flow Attention.** Leveraging the rotation-covariant descriptors $\mathbf{Q}_{\text{hist}}$ and $\mathbf{Q}_{\text{flow}}^\tau$, we design an attention mechanism to evaluate the cross-temporal compatibility between historical context and future flow. To ensure strict invariance, we measure the affinity via the scalar product over the feature dimension. Specifically, for each node $n$, we compute the attention score between the historical frame $t_h$ and the future frame $t_f$ as:

$$\mathbf{S}^\tau(n, t_h, t_f) = \big\langle \mathbf{Q}_{\text{hist}}(n, t_h), \mathbf{Q}_{\text{flow}}^\tau(n, t_f) \big\rangle, \tag{10}$$

where $\langle \cdot, \cdot \rangle$ is standard inner product in $\mathbb{R}^{3D}$. This operation contracts the covariant feature dimensions, yielding an invariant attention map $\mathbf{S}^\tau \in \mathbb{R}^{N \times T_h \times T_f}$. The normalized attention weight is then obtained via temporal softmax:

$$\mathbf{A}^\tau(n, t_h, t_f) = \frac{\exp\big(\mathbf{S}^\tau(n, t_h, t_f)\big)}{\sum_{k=1}^{T_h} \exp\big(\mathbf{S}^\tau(n, k, t_f)\big)}. \tag{11}$$

**Equivariant Trajectory Integration.** To complement instantaneous flow predictions with long-term structural dependencies, we propose a geometric integration strategy. First, we compute the relative geometric embeddings $\Delta\mathbf{C}_X$, anchored at the last observed frame $T_h$, to isolate intrinsic historical geometry patterns:

$$\Delta\mathbf{C}_X(n, t_h) = \mathbf{C}_X(n, t_h) - \mathbf{C}_X(n, T_h). \tag{12}$$

We then synthesize a history-inferred state $\mathbf{Y}_{\text{ctx}}^\tau$ by aggregating these relative embeddings via attention $\mathbf{A}^\tau$ and combining them onto the current anchor:

$$\mathbf{Y}_{\text{ctx}}^\tau(n, t_f) = \mathbf{C}_X(n, T_h) + \sum_{t_h=1}^{T_h} \mathbf{A}^\tau(n, t_h, t_f) \cdot \Delta\mathbf{C}_X(n, t_h), \tag{13}$$

Finally, the target trajectory is obtained by fusing this history-inferred state with the flow-based prediction $\hat{\mathbf{Y}}^\tau$:

$$\mathbf{Y}_{\text{pred}}^\tau = \frac{\hat{\mathbf{Y}}^\tau + \mathbf{Y}_{\text{ctx}}^\tau}{2}. \tag{14}$$

This design refines the flow prediction with long-term geometric coherence derived from the historical context. The rotation-equivariance of the output $\mathbf{Y}_{\text{pred}}^\tau \in \mathbb{R}^{N \times T_f \times 3}$ is guaranteed by the following lemma.

**Lemma 3.2.** *The attention weight $\mathbf{A}^\tau$ in Eq. (11) is rotation-invariant, and the generated trajectory $\mathbf{Y}_{pred}^\tau$ in Eq. (14) is rotation-equivariant with respect to input position $\mathbf{X}$.*

### 3.5. Network Training and Inference

The proposed network $\boldsymbol{\Phi}$ integrates the HCE, FSE, and TG modules to map noisy state to clean trajectory. During *Training*, we optimize $\boldsymbol{\Phi}$ via the flow matching objective in Eq. (2). By regressing the clean target $\mathbf{Y}_{\text{pred}}^\tau$, the model explicitly aligns the generative flow with the ground truth data distribution. During *Inference*, we synthesize the future trajectory by numerically integrating the estimated instantaneous velocity as in Eq. (4). Detailed algorithmic procedures for training and inference are provided in the Appendix B.

We formally establish the geometric consistency of the proposed framework in the following theorem:

**Theorem 3.3.** *The entire generative process of GSE-Flow, initialized with isotropic Gaussian noise and evolved via the learned equivariant flow, yields an SE(3)-equivariant trajectory with respect to the original input observation $\mathbf{P}$.*

Lemmas 3.1–3.2 and Theorem 3.3 guarantee equivariance of the generation pipeline from initialization to integration. Detailed proofs are provided in Appendix A.

## 4. Experiments

We evaluate GSE-Flow across diverse spatiotemporal scales for geometric trajectory prediction, ranging from small molecules (§4.1) and macromolecules (§4.2) to macroscopic human motion (§4.3), with ablation analysis in §4.5.

**Datasets.** We evaluate GSE-Flow on three benchmarks: MD17 (Chmiela et al., 2017), MD22 (Chmiela et al., 2023), and CMU MoCap (CMU, 2003). (1) **MD17** consists of trajectories for eight small organic molecules ranging from 9 to 21 atoms. Input features comprise atomic numbers as SE(3)-invariant scalars and Cartesian coordinates as geometric states. (2) **MD22** includes seven macromolecules with 42 to 370 atoms serving as a scalability testbed. While sharing the same input representation as MD17, these systems exhibit significantly higher geometric complexity. (3) **CMU MoCap** captures human kinematics, specifically *Walk* (Subject #35), characterized by regular, periodic locomotion, and *Basketball* (Subject #102), featuring highly dynamic, non-linear maneuvers. Across all datasets, we adopt the long-term forecasting setting from EqMotion (Xu et al., 2023) to predict 10 future frames given 10 historical frames.

**Evaluation Metrics.** All models are evaluated using Av-

*Table 1.* Results (ADE/FDE, $\times 10^{-2}$) on MD17. **Bold** and underlined indicate the best and second-best performance, respectively.

| Method | Aspirin | Benzene | Ethanol | Malonaldehyde | Naphthalene | Salicylic | Toluene | Uracil |
|---|---|---|---|---|---|---|---|---|
| TFN(Thomas et al., 2018) | 11.84/17.17 | 6.30/8.98 | 6.55/8.20 | 10.61/15.98 | 9.06/11.10 | 10.49/14.27 | 7.39/8.21 | 9.42/11.89 |
| SE(3)-TR(Fuchs et al., 2020) | 12.07/16.96 | 6.35/9.03 | 6.50/7.98 | 10.87/16.00 | 8.71/10.58 | 10.37/13.62 | 6.86/7.09 | 9.27/11.38 |
| EGNN(Satorras et al., 2021) | 10.13/15.12 | 4.81/6.68 | 5.16/7.10 | 7.09/10.88 | 7.37/9.65 | 9.77/13.98 | 5.83/7.05 | 5.61/7.34 |
| FAGNN(Puny et al., 2022) | 6.92/9.44 | 1.54/2.32 | 4.84/6.80 | **5.48/8.13** | 4.13/4.60 | 4.80/**5.85** | 3.77/4.38 | 3.81/**4.42** |
| EGHN(Han et al., 2022) | 9.11/13.83 | 3.32/5.64 | 5.05/7.13 | 7.36/11.36 | 6.54/8.13 | 9.79/12.89 | 5.72/6.95 | 6.02/7.06 |
| EqMotion(Xu et al., 2023) | 7.05/10.62 | **1.30/2.27** | 4.61/**6.54** | 6.33/9.57 | 3.59/4.83 | 4.53/6.37 | 3.13/4.24 | 3.61/4.85 |
| ESTAG(Wu et al., 2023) | 6.77/10.87 | 3.40/5.95 | 4.83/6.88 | 6.53/10.09 | 6.09/7.97 | 8.08/12.98 | 5.49/7.41 | 4.83/6.22 |
| EGNO(Xu et al., 2024) | 7.27/12.18 | 3.35/5.21 | 5.11/7.11 | 6.64/10.55 | 6.14/8.76 | 6.62/11.18 | 5.59/7.61 | 5.09/7.48 |
| GeoTDM(Han et al., 2024) | 14.92/22.70 | 4.33/8.84 | 7.78/10.56 | 14.21/21.97 | 8.75/10.43 | 13.78/19.90 | 10.41/12.88 | 9.80/12.35 |
| PT-EvNet(Yu & Sun, 2024) | 6.21/9.32 | 1.47/2.64 | 4.65/6.74 | 5.74/10.04 | 4.32/5.04 | 4.63/6.44 | 5.01/6.11 | 3.77/4.74 |
| FMN(Huang et al., 2025) | 8.32/11.72 | 2.27/3.72 | 6.03/7.79 | 7.83/11.64 | 4.20/4.91 | 5.21/6.32 | 4.95/5.89 | 4.60/5.29 |
| Ours | **4.85/7.96** | 1.39/2.49 | **4.48/**6.60 | 5.68/9.17 | **3.12/4.34** | **4.09/**6.01 | **3.09/4.17** | **3.43/**4.61 |

*Table 2.* Trajectory prediction results (ADE/FDE, $\times 10^{-2}$) on MD22. Lower is better.

| Method | AT-AT | AT-AT-CG-CG | Ac-Ala$_3$-NHMe | DHA | Buckyball-catcher | Double-walled nanotube | Stachyose |
|---|---|---|---|---|---|---|---|
| TFN(Thomas et al., 2018) | 22.01/35.05 | 24.77/40.66 | 22.61/35.76 | 23.03/37.00 | 16.23/24.44 | 19.84/28.12 | 21.99/33.33 |
| SE(3)-TR(Fuchs et al., 2020) | 22.12/34.60 | 24.42/39.59 | 22.46/34.68 | 23.35/36.66 | 15.54/22.90 | 18.81/25.70 | 21.77/31.87 |
| EGNN(Satorras et al., 2021) | 21.73/33.46 | 22.47/37.48 | 21.65/34.89 | 20.45/33.23 | 16.07/23.96 | 19.65/28.14 | 20.87/32.30 |
| FAGNN(Puny et al., 2022) | 13.62/20.90 | 16.45/26.07 | 14.99/22.49 | 17.14/26.74 | 12.67/17.26 | 13.68/17.96 | 17.65/26.93 |
| EGHN(Han et al., 2022) | 15.67/25.10 | 16.40/27.44 | 16.87/26.58 | 16.64/27.60 | 13.02/18.35 | 16.91/22.07 | 19.59/30.29 |
| EqMotion(Xu et al., 2023) | 11.59/18.37 | 12.35/19.86 | 11.90/19.07 | 13.65/22.60 | 15.34/21.43 | 543.58/436.66 | 15.31/24.48 |
| ESTAG(Wu et al., 2023) | 14.05/23.10 | 15.01/25.19 | 14.24/23.11 | 15.45/25.99 | 11.24/15.91 | 13.67/19.21 | 17.52/27.61 |
| EGNO(Xu et al., 2024) | 13.58/23.51 | 14.46/25.30 | 15.77/26.79 | 15.67/27.30 | 12.21/18.54 | 15.50/21.87 | 17.71/28.95 |
| GeoTDM(Han et al., 2024) | 27.22/41.76 | 26.25/42.09 | 28.05/43.44 | 28.28/44.07 | 20.79/28.40 | 25.46/35.23 | 29.20/44.49 |
| PT-EvNet(Yu & Sun, 2024) | 12.14/20.71 | 14.37/22.46 | 11.55/22.31 | 14.77/24.36 | 10.91/13.42 | OOM | 17.74/26.44 |
| FMN(Huang et al., 2025) | 14.61/21.74 | 14.13/21.42 | 16.44/24.79 | 17.28/27.32 | 9.66/**11.48** | OOM | 17.53/25.48 |
| Ours | **9.15/16.32** | **10.35/19.09** | **10.94/19.01** | **11.45/20.86** | **7.52**/12.20 | **10.26/15.70** | **14.66/24.39** |

erage Displacement Error (ADE) and Final Displacement Error (FDE), which measure the $\ell_2$ distance between predicted and ground truth trajectories averaged across all future frames and at the final endpoint, respectively.

To ensure fair comparison, we reproduced the results for all compared methods based on their official codes. Further details on the dataset, experimental setup and baseline configurations are provided in Appendix C. The source code and processed datasets will be made publicly available.

### 4.1. Results on MD17

**Implementation Details.** For MD17, we utilize particle coordinates as input absolute positions $\mathbf{P}$ and atomic numbers as invariant attributes $\mathbf{H}$. We construct the graph by connecting atoms within a predefined distance threshold as 1-hop neighbors. To capture broader interactions, the final connectivity $\mathbf{E}$ incorporates 1-hop and 2-hop neighbors.

**Results and Comparisons.** Table 1 presents the quantitative evaluation on the MD17 benchmark. GSE-Flow demonstrates superior predictive fidelity, achieving the lowest ADE

on 6 out of 8 molecular systems. Notably, on Aspirin, which is characterized by complex dynamics, our method yields an approximate 22% reduction in ADE relative to the best-performing baseline. While maintaining performance parity with leading methods on smaller systems such as Benzene, GSE-Flow establishes a pronounced advantage on larger structures like Naphthalene and Salicylic. These results substantiate the efficacy of the proposed framework in modeling long-term geometric trajectories.

### 4.2. Results on MD22

**Implementation Details.** To assess the effectiveness and scalability of GSE-Flow, we apply the identical experimental configuration (inputs and connectivity construction) used for MD17 to the larger macromolecules in MD22. This uniform setup verifies the model's capacity to adapt to increased geometric complexity.

**Results and Comparisons.** As presented in Table 2, GSE-Flow achieves the best ADE across all seven subsets. Quantitatively, our method outperforms the second-best approaches by a substantial margin, delivering a relative

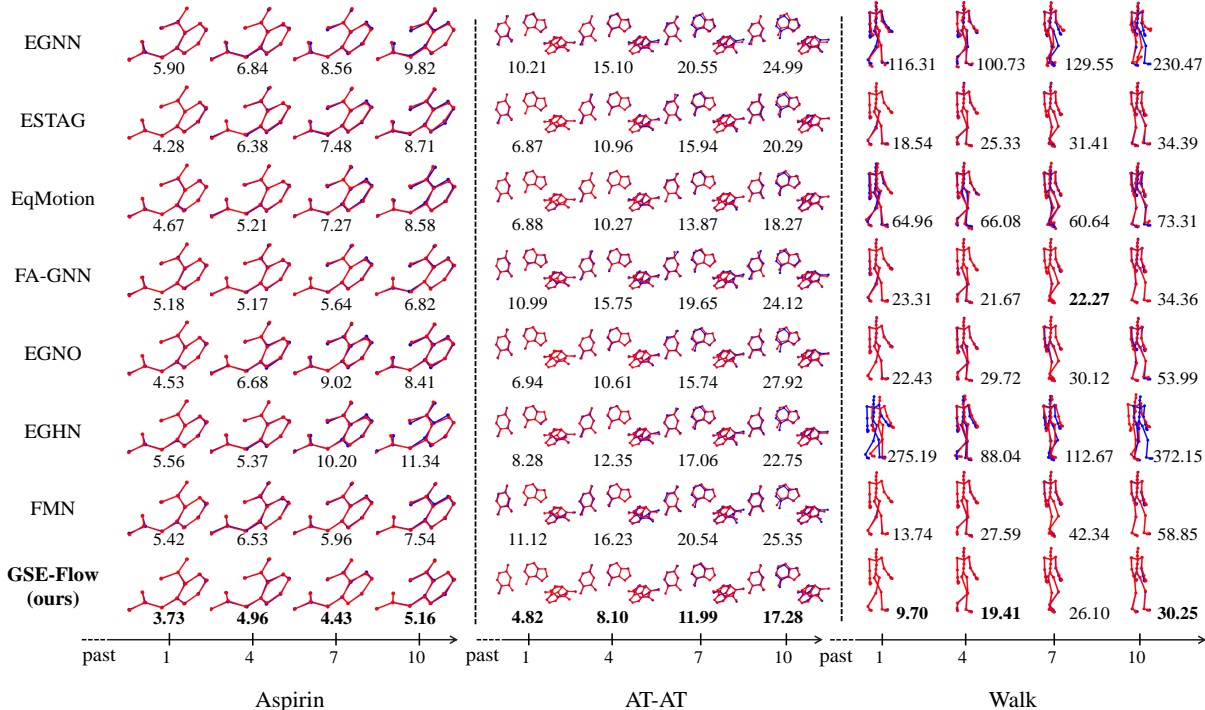

*Figure 2.* Visualization of trajectory predictions. The ground truth is in blue, and predictions are in red. The numbers below each snapshot indicate the prediction error ($\ell_2$ distance, $\times 10^{-2}$). GSE-Flow achieves lower errors and maintains better structural consistency across datasets of varying scales, even over long-term horizons. Please zoom in for better visualization.

improvement of up to ~25% on complex structures like the Double-walled nanotube. For FDE, our model ranks first in 6 out of 7 cases. Crucially, GSE-Flow demonstrates superior stability and efficiency, avoiding numerical divergence observed in EqMotion (Xu et al., 2023) and out-of-memory (OOM) failures encountered by methods that explicitly increase the number of coordinate systems such as PT-EvNet (Yu & Sun, 2024) and FMN (Huang et al., 2025).

### 4.3. Results on CMU MoCap

**Implementation Details.** For the CMU MoCap dataset, we define the input absolute position $\mathbf{P}$ using joint coordinates and initialize node attributes as constant scalars ($\mathbf{H} = \mathbf{1}$). Leveraging the fixed skeletal topology, we construct the connectivity $\mathbf{E}$ by incorporating both 1-hop and 2-hop joint neighbors to capture broader kinematic dependencies.

**Results and Comparisons.** Table 3 details the quantitative comparisons. GSE-Flow demonstrates robust performance across distinct dynamic scenarios. On the Walk subset, our method achieves state-of-the-art accuracy in both ADE and FDE, outperforming all baselines. In the more challenging Basketball scenario, GSE-Flow attains the lowest ADE and the second-best FDE. Overall, these results confirm that GSE-Flow can robustly model complex dynamics, preserving high fidelity in both highly structured gaits and

*Table 3.* Performance of ADE / FDE ($\times 10^{-1}$) on CMU MoCap.

| Method | Walk | Basketball |
|---|---|---|
| TFN(Thomas et al., 2018) | 9.41/11.8 | 103.11/165.93 |
| SE(3)-TR(Fuchs et al., 2020) | 54.3/62.4 | 123.72/147.64 |
| EGNN(Satorras et al., 2021) | 7.22/10.80 | 99.20/154.53 |
| FAGNN(Puny et al., 2022) | 3.12/4.42 | 56.32/78.41 |
| EGHN(Han et al., 2022) | 6.25/9.11 | 39.94/68.06 |
| EqMotion(Xu et al., 2023) | 5.85/7.52 | 33.67/**56.46** |
| ESTAG(Wu et al., 2023) | 3.27/5.20 | 38.85/67.49 |
| EGNO(Xu et al., 2024) | 3.63/6.06 | 38.98/73.81 |
| GeoTDM(Han et al., 2024) | 16.66/32.32 | 63.70/108.13 |
| PT-EvNet(Yu & Sun, 2024) | 3.24/4.91 | 37.21/76.24 |
| FMN(Huang et al., 2025) | 3.03/4.88 | 45.99/79.50 |
| Ours | **2.59/4.25** | **32.39**/57.63 |

unpredictable, fast-paced movements.

### 4.4. Visualization

Figure 2 compares the trajectory predictions of GSE-Flow and representative baselines on the Aspirin, AT-AT, and Walk datasets at steps $t = \{1, 4, 7, 10\}$. Ground truth and predicted states are shown in blue and red, with $\ell_2$ errors marked below. As shown, most methods work well on the simpler Aspirin molecule and yield accurate estimates in the initial frames. However, the increased structural complexity of the AT-AT structure and Walk dynamics leads to notable

*Table 4.* Ablation study results (ADE / FDE). * denotes GSE-Flow.

|  | Ben.($\times 10^{-2}$) | Walk($\times 10^{-1}$) | AT-AT($\times 10^{-2}$) |
|---|---|---|---|
| 1) *-w/o-hist | 2.35/4.08 | 85.84/158.90 | 29.85/38.37 |
| 2) *-w/o-MP | 1.93/3.45 | 26.66/43.52 | 10.00/17.35 |
| 3) *-w/o-MV | 1.43/2.61 | 27.25/45.14 | 10.95/20.05 |
| 4) *-w/o-MF | 1.43/2.56 | 27.69/46.08 | 10.81/18.40 |
| 5) *-GA | 1.67/2.97 | 26.19/43.35 | 9.80/17.13 |
| 6) *-FA | 1.80/3.15 | 25.93/43.23 | 10.05/17.04 |
| 7) *- GA+FA | 1.59/2.82 | 26.99/43.45 | 9.64/17.60 |
| 8) *-AG | 1.45/2.71 | 26.37/43.76 | 9.34/16.07 |
| 9) *-w/o-Res | 4.07/6.54 | 28.63/48.23 | 14.97/24.82 |
| 10) *-VF | 2.50/3.97 | 30.90/49.46 | 12.77/23.11 |
| GSE-Flow | **1.39/2.49** | **25.92/42.54** | **9.15/16.32** |

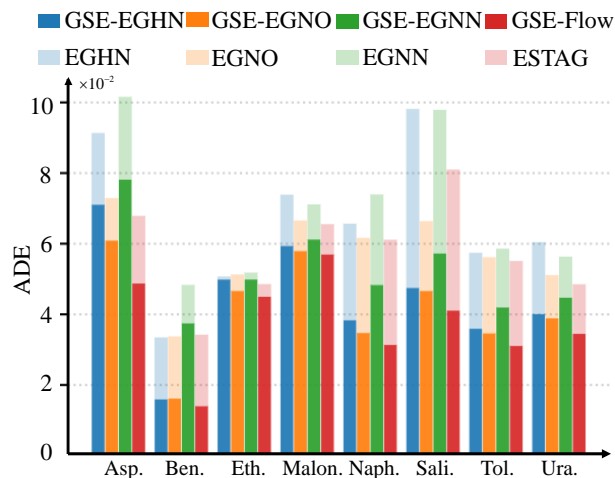

*Figure 3.* ADE ($\times 10^{-2}$) results for deterministic baselines (faded bars) and their GSE-Flow integrated counterparts (solid bars).

performance degradation in baseline models. As the prediction horizon extends, these methods tend to diverge from the ground truth, exhibiting increased spatial deviation and error accumulation by the 10-th frame. In contrast, GSE-Flow maintains consistent alignment with the target trajectories. These qualitative observations, consistent with quantitative results in Sec. 4.1–4.3, demonstrate that our model effectively preserves geometric structure across different scales, ranging from molecular conformations to human skeletal dynamics, ensuring high fidelity even at later frames.

### 4.5. Ablation Studies

To evaluate the efficacy of our specific designs and verify the generality of the proposed architecture, we conduct ablation studies on the Benzene subset from MD17, the AT-AT subset from MD22, and the Walk subset from CMU MoCap.

**Effectiveness of Coherent Sequence Encoding.** We validate the integration of historical context in the FSE module (Eq. (6)) by evaluating **1) GSE-Flow-w/o-hist**, a variant encoding only the noisy flow state. The significant performance drop in Table 4 confirms that incorporating historical cues is critical for capturing temporal dependencies.

**Effectiveness of Time-Modulated Embedding.** To validate the time injection mechanism, we evaluate **2) GSE-Flow-w/o-MP** and **3) GSE-Flow-w/o-MV**, which exclude the position and velocity modulation terms in Eq. (7), respectively, alongside **4) GSE-Flow-w/o-MF** for the node feature modulation in Eq. (8). The consistent performance decline in Table 4 confirms that explicitly modulating all state representations with flow time is essential.

**Effectiveness of Geometry-Feature Tensorization.** To assess the fusion strategy based on geometry-feature tensorization (Eq. (10)) in the TG module, we compare it with **5) GSE-Flow-GA** using attention scores from only geometric position, **6) GSE-Flow-FA** using only node features, and **7) GSE-Flow-GA+FA** summing their scores. The results in Table 4 demonstrates that our design is more effective than single-modality or additive fusion.

**Effectiveness of Relative Displacement Aggregation.** To validate the relative displacement strategy (Eqs. (12–13)), we compare it with **8) GSE-Flow-AG** by aggregating absolute representations $\mathbf{C}_X$. Table 4 confirms that relative context consistently outperforms absolute aggregation.

**Effectiveness of Residual Connection.** To validate the residual connection (Eq. (14)), we evaluate **9) GSE-Flow-w/o-Res** without it. The performance drop confirms that pure attention restricts predictions to the historical linear span, which is insufficient for capturing complex dynamics.

**Effectiveness of Trajectory Prediction Strategy.** In GSE-Flow, we derive vector fields from predicted trajectories (Eq. (3)) rather than predicting them directly. We compare this with **10) GSE-Flow-VF**, which directly predicts instantaneous vector fields. The performance gap in Table 4 validates the superiority of our trajectory-driven strategy.

**Generality of GSE-Flow.** We validate the generality of our method by integrating GSE-Flow with deterministic baselines (EGNN, EGHN, EGNO) via the substitution of sequence encoders in Eqs. (5) and (6). This integration yields consistent gains on MD17 as in Figure 3, proving that our probabilistic flow matching effectively enhances deterministic backbones. Detailed quantitative results for MD17, MD22, and CMU MoCap are included in Appendix E.

## 5. Limitations

GSE-Flow instantiates conditional-flow modeling for trajectory prediction, but empirical evidence is still anchored in point-wise displacement metrics (ADE/FDE), and systematic exploration of multimodal coverage, calibration, and likelihood-style diagnostics remains comparatively limited.

Compositionally the design builds on established equivariant components with standard conditional-flow objectives, so algebraic equivariance tracks the chosen backbone; sharpening a principled perspective on how discrete history constrains continuous future motion remains a natural direction to advance from the same framework.

## 6. Conclusion

In this work, we present GSE-Flow, a framework that advances SE(3)-equivariant trajectory prediction from deterministic regression to a continuous generative paradigm. Our approach explicitly addresses the challenge of conditioning stochastic flow dynamics on geometric history. Specifically, we achieve this through Coherent Sequence Encoding to unify temporal streams, Time-Modulated Embeddings to inject evolution time into geometric states, and Context-Flow Fusion to align generative predictions with structural priors. Empirical results on MD17, MD22, and CMU MoCap confirm that GSE-Flow yields SOTA accuracy and effectively enhances deterministic backbones. Future work will extend this equivariant generative paradigm to large-scale molecular simulations and embodied planning.

## Acknowledgements

This work was supported by NSFC with grant numbers of 62306167, 12125104, 12426313, and project ZR2024QA161 supported by Shandong Provincial Natural Science Foundation. It was also funded by National Social Science Fund Project (25CJY023), and Key-Area Research and Development Program of Guangdong Province (No.2025B1111120001).

## Impact Statement

This paper presents work whose goal is to advance the field of Machine Learning. There are many potential societal consequences of our work, none which we feel must be specifically highlighted here.

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

# A. Proofs

We provide proofs for Lemmas 3.1–3.2 and Theorem 3.3.

**Preliminaries.** We consider the Special Euclidean group $\mathrm{SE}(3)$, where a group element $g = (\mathbf{R}, \mathbf{t})$ consists of a rotation matrix $\mathbf{R} \in \mathrm{SO}(3)$ and a translation vector $\mathbf{t} \in \mathbb{R}^3$. A function $\mathcal{F}$ is defined as $\mathrm{SE}(3)$-*equivariant* if transforming the input $\mathbf{P}$ results in a corresponding transformation of the output:

$$T_g\big(\mathcal{F}(\mathbf{P})\big) = \mathcal{F}\big(T_g(\mathbf{P})\big), \tag{15}$$

where $T_g$ denotes the transformation operator induced by $g$, defined as $T_g(\mathbf{P}) = \mathbf{RP} + \mathbf{t}$.

**Translation Normalization.** To ensure translation equivariance, we employ the translation normalization strategy by subtracting the spatiotemporal centroid from the input sequence. Let $\overline{\mathbf{P}}$ denote the centroid of the historical positions. The spatiotemporally centered relative coordinate $\mathbf{X}$ is defined as:

$$\mathbf{X} = \mathbf{P} - \overline{\mathbf{P}}, \quad \text{where} \quad \overline{\mathbf{P}} = \frac{1}{N \cdot T_h} \sum_{i=1}^{N} \sum_{t=1}^{T_h} \mathbf{P}_{i,t}. \tag{16}$$

Crucially, under a global transformation of the original data $\mathbf{P}' = \mathbf{RP} + \mathbf{t}$, the centroid transforms consistently as $\overline{\mathbf{P}}' = \mathbf{R}\overline{\mathbf{P}} + \mathbf{t}$. Consequently, the transformed relative coordinate $\mathbf{X}'$ is derived as:

$$\begin{aligned}
\mathbf{X}' &= \mathbf{P}' - \overline{\mathbf{P}}' \\
&= (\mathbf{RP} + \mathbf{t}) - (\mathbf{R}\overline{\mathbf{P}} + \mathbf{t}) \\
&= \mathbf{R}(\mathbf{P} - \overline{\mathbf{P}}) \\
&= \mathbf{RX}.
\end{aligned} \tag{17}$$

Equation (17) demonstrates that the translation $\mathbf{t}$ is eliminated in the centered coordinate $\mathbf{X}$. Thus, the problem of learning an $\mathrm{SE}(3)$-equivariant function on $\mathbf{P}$ reduces to learning an $\mathrm{SO}(3)$-equivariant function on $\mathbf{X}$. The final prediction is recovered by adding the original centroid $\overline{\mathbf{P}}$ back to the model output. Given this decoupling, our subsequent analysis focuses exclusively on establishing equivariance with respect to the rotation group $\mathrm{SO}(3)$.

## A.1. Proof of Lemma 3.1

**Lemma 3.1.** *The time-modulated position $\hat{\mathbf{Y}}^\tau$ in Eq. (7) is rotation-equivariant with respect to the input position $\mathbf{X}$.*

*Proof.* Given the translation normalization strategy, the translation component is explicitly eliminated. Thus, it suffices to analyze equivariance with respect to the rotation group $\mathrm{SO}(3)$. Let $T_\mathbf{R}$ denote the action of a rotation matrix $\mathbf{R} \in \mathrm{SO}(3)$. We present the proof in three steps by verifying the $\mathrm{SO}(3)$-equivariance of the flow input construction, the backbone processing by $\psi$, and the time modulation.

**Step 1: Equivariance of Flow Input Construction.** The geometric input to the FSE module consists of the historical observation $\mathbf{X}$ and the noisy flow state $\mathbf{Y}^\tau$. The flow state is constructed via linear interpolation: $\mathbf{Y}^\tau = (1 - \tau)\mathbf{Y}^0 + \tau\mathbf{Y}^1$, where $\mathbf{Y}^1$ represents the ground truth geometric coordinates and $\mathbf{Y}^0$ is the sampled Gaussian noise. Under a global rotation $\mathbf{R}$, the geometric vectors transform as:

$$T_\mathbf{R}(\mathbf{X}) = \mathbf{RX}, \quad T_\mathbf{R}(\mathbf{Y}^1) = \mathbf{RY}^1. \tag{18}$$

Since $\mathbf{Y}^0$ is treated as a geometric vector field in the centered frame, it transforms consistently: $T_\mathbf{R}(\mathbf{Y}^0) = \mathbf{RY}^0$. Consequently, the interpolated input $\mathbf{Y}^\tau$ preserves this linearity:

$$T_\mathbf{R}(\mathbf{Y}^\tau) = (1 - \tau)\mathbf{RY}^0 + \tau\mathbf{RY}^1 = \mathbf{RY}^\tau. \tag{19}$$

**Step 2: Equivariance of Backbone Processing.** The backbone $\psi$ takes the concatenated sequence $[\mathbf{X}, \mathbf{Y}^\tau]$ as input. Since $\psi$ is instantiated as an $\mathrm{SO}(3)$-equivariant architecture (adapted from ESTAG (Wu et al., 2023)), its output transforms equivariantly with the input. Let $\tilde{\mathbf{Y}}^\tau$ denote the output slice corresponding to the flow state (Eq. (6)). We have:

$$T_\mathbf{R}(\tilde{\mathbf{Y}}^\tau) = \mathrm{Slice}\left(\psi(T_\mathbf{R}([\mathbf{X}, \mathbf{Y}^\tau]), \dots)\right) = \mathrm{Slice}\left(\psi([\mathbf{RX}, \mathbf{RY}^\tau], \dots)\right). \tag{20}$$

By the equivariance property of $\psi$, the rotation factorizes out:

$$T_{\mathbf{R}}(\tilde{\mathbf{Y}}^\tau) = \mathbf{R} \cdot \text{Slice}\left(\psi([\mathbf{X}, \mathbf{Y}^\tau], \dots)\right) = \mathbf{R}\tilde{\mathbf{Y}}^\tau. \tag{21}$$

**Step 3: Equivariance of Time Modulation.** Finally, consider the time-modulated output $\hat{\mathbf{Y}}^\tau = (1 + \gamma_{eq})\tilde{\mathbf{Y}}^\tau + \beta_{eq}\tilde{\mathbf{V}}^\tau$ (Eq. (7)). The velocity $\tilde{\mathbf{V}}^\tau$ is derived from $\hat{\mathbf{Y}}^\tau$ via temporal finite differences. Since rotation is a linear operator, it commutes with the difference operator $\Delta_t$:

$$T_{\mathbf{R}}(\tilde{\mathbf{V}}^\tau) = T_{\mathbf{R}}(\Delta_t \tilde{\mathbf{Y}}^\tau) = \Delta_t(\mathbf{R}\tilde{\mathbf{Y}}^\tau) = \mathbf{R}\tilde{\mathbf{V}}^\tau. \tag{22}$$

As the scalars $\gamma_{eq}, \beta_{eq}$ are derived solely from $\tau$ and are SO(3)-invariant, we obtain:

$$T_{\mathbf{R}}(\hat{\mathbf{Y}}^\tau) = (1 + \gamma_{eq})\mathbf{R}\tilde{\mathbf{Y}}^\tau + \beta_{eq}\mathbf{R}\tilde{\mathbf{V}}^\tau = \mathbf{R}\hat{\mathbf{Y}}^\tau. \tag{23}$$

Combining these steps, we conclude that $\hat{\mathbf{Y}}^\tau$ is rotation-equivariant with respect to the input position $\mathbf{X}$. $\qquad\square$

### A.2. Proof of Lemma 3.2

**Lemma 3.2.** *The attention weight $\mathbf{A}^\tau$ in Eq. (11) is rotation-invariant, and the generated trajectory $\mathbf{Y}^\tau_{pred}$ in Eq. (14) is rotation-equivariant with respect to input position $\mathbf{X}$.*

*Proof.* As established in the preliminaries and translation normalization, we operate in the centered coordinate frame where translation is explicitly normalized. Consequently, the translation component is eliminated from the input representation. Thus, our proof focuses on establishing invariance and equivariance under the rotation group SO(3). Let $T_{\mathbf{R}}$ denote the action of a rotation matrix $\mathbf{R} \in SO(3)$.

**Step 1: Rotation-Invariance of Attention Weight $\mathbf{A}^\tau$.** Recall that the geometry-feature tensor is defined as $\mathbf{Q}_{\text{hist}} = \text{vec}(\mathbf{C}_X \otimes \mathbf{C}_H)$, where $\mathbf{C}_X$ and $\mathbf{C}_H$ are the equivariant and invariant representations from the HCE module, respectively. Under a rotation $\mathbf{R}$, the geometric component transforms as $\mathbf{C}'_X = \mathbf{R}\mathbf{C}_X$, while the invariant feature $\mathbf{C}_H$ remains unchanged. Utilizing the properties of the vectorization operator, the tensor transforms as:

$$\mathbf{Q}'_{\text{hist}} = \text{vec}((\mathbf{R}\mathbf{C}_X) \otimes \mathbf{C}_H) = (\mathbf{I}_D \otimes \mathbf{R})\text{vec}(\mathbf{C}_X \otimes \mathbf{C}_H) = \tilde{\mathbf{R}}\mathbf{Q}_{\text{hist}}, \tag{24}$$

where $\tilde{\mathbf{R}} = (\mathbf{I}_D \otimes \mathbf{R})$ is a block-diagonal rotation matrix satisfying orthogonality $\tilde{\mathbf{R}}^\top\tilde{\mathbf{R}} = \mathbf{I}$. Analogously, invoking Lemma 3.1, the flow geometric state $\hat{\mathbf{Y}}^\tau$ is SO(3)-equivariant while the feature $\hat{\mathbf{Z}}^\tau$ is invariant. Consequently, the flow tensor exhibits an identical transformation property: $\mathbf{Q}^\tau_{\text{flow}}{}' = \tilde{\mathbf{R}}\mathbf{Q}^\tau_{\text{flow}}$.

The affinity score $\mathbf{S}^\tau$ is computed via the inner product. Under rotation, it transforms as:

$$\mathbf{S}^{\tau\prime} = \left\langle \mathbf{Q}'_{\text{hist}}, \mathbf{Q}^\tau_{\text{flow}}{}' \right\rangle = (\mathbf{Q}'_{\text{hist}})^\top \mathbf{Q}^\tau_{\text{flow}}{}' = (\tilde{\mathbf{R}}\mathbf{Q}_{\text{hist}})^\top(\tilde{\mathbf{R}}\mathbf{Q}^\tau_{\text{flow}}) = \mathbf{Q}^\top_{\text{hist}}(\tilde{\mathbf{R}}^\top\tilde{\mathbf{R}})\mathbf{Q}^\tau_{\text{flow}} = \mathbf{S}^\tau. \tag{25}$$

Since the affinity score $\mathbf{S}^\tau$ is invariant to rotation, the softmax operation in Eq. (11) preserves this property. Thus, the attention weights $\mathbf{A}^\tau$ are SO(3)-invariant.

**Step 2: SO(3)-Equivariance of Context Trajectory $\mathbf{Y}^\tau_{\text{ctx}}$.** The relative geometric embedding is defined as $\Delta\mathbf{C}_X(n, t_h) = \mathbf{C}_X(n, t_h) - \mathbf{C}_X(n, T_h)$. Since $\mathbf{C}_X$ is SO(3)-equivariant (in the centered frame), the difference operator preserves this equivariance:

$$T_{\mathbf{R}}(\Delta\mathbf{C}_X(n, t_h)) = \mathbf{R}\mathbf{C}_X(n, t_h) - \mathbf{R}\mathbf{C}_X(n, T_h) = \mathbf{R}\Delta\mathbf{C}_X(n, t_h). \tag{26}$$

The context trajectory $\mathbf{Y}^\tau_{\text{ctx}}$ is a linear combination of these equivariant embeddings weighted by the invariant attention $\mathbf{A}^\tau$, added to the equivariant anchor $\mathbf{C}_X(n, T_h)$. Its transformation is given by:

$$
\begin{aligned}
T_{\mathbf{R}}(\mathbf{Y}^\tau_{\text{ctx}}(n, t_f)) &= \mathbf{R}\mathbf{C}_X(n, T_h) + \sum_{t_h=1}^{T_h} \mathbf{A}^\tau(n, t_h, t_f) \cdot \mathbf{R}\Delta\mathbf{C}_X(n, t_h) \\
&= \mathbf{R}\left(\mathbf{C}_X(n, T_h) + \sum_{t_h=1}^{T_h} \mathbf{A}^\tau(n, t_h, t_f) \cdot \Delta\mathbf{C}_X(n, t_h)\right) \\
&= \mathbf{R}\mathbf{Y}^\tau_{\text{ctx}}(n, t_f).
\end{aligned}
\tag{27}
$$

Thus, $\mathbf{Y}_{\text{ctx}}^{\tau}$ is SO(3)-equivariant.

**Step 3: SO(3)-Equivariance of Final Prediction $\mathbf{Y}_{\text{pred}}^{\tau}$.** From Lemma 3.1, the flow-modulated state $\hat{\mathbf{Y}}^{\tau}$ is SO(3)-equivariant. Since $\mathbf{Y}_{\text{ctx}}^{\tau}$ is also equivariant (Step 2), their linear combination in Eq. (14) preserves equivariance:

$$T_{\mathbf{R}}(\mathbf{Y}_{\text{pred}}^{\tau}) = \frac{T_{\mathbf{R}}(\hat{\mathbf{Y}}^{\tau}) + T_{\mathbf{R}}(\mathbf{Y}_{\text{ctx}}^{\tau})}{2} = \frac{\mathbf{R}\hat{\mathbf{Y}}^{\tau} + \mathbf{R}\mathbf{Y}_{\text{ctx}}^{\tau}}{2} = \mathbf{R}\mathbf{Y}_{\text{pred}}^{\tau}. \tag{28}$$

$\square$

### A.3. Proof of Theorem 3.3

**Theorem 3.3.** *The entire generative process of GSE-Flow, initialized with isotropic Gaussian noise and evolved via the learned equivariant flow, yields an SE(3)-equivariant trajectory with respect to the original input observation* $\mathbf{P}$.

*Proof.* Given the translation normalization defined in the Preliminaries, the generative flow operates in the centered coordinate system. We structure the proof in two parts: first proving the SO(3)-equivariance of the relative trajectory generation via induction, and then establishing the SE(3)-equivariance of the final absolute trajectory.

**Part 1: SO(3)-Equivariance of Relative Flow.** Let $T_{\mathbf{R}}$ denote the rotation operator such that $T_{\mathbf{R}}(\mathbf{X}) = \mathbf{R}\mathbf{X}$. We examine the iterative evolution of the relative state $\mathbf{Y}^{\tau}$.

*Base Case ($\tau = 0$):* The process initializes with $\mathbf{Y}^0 \sim \mathcal{N}(\mathbf{0}, \mathbf{I})$. Under a rotation of the input observation, the isotropic noise sample transforms as $T_{\mathbf{R}}(\mathbf{Y}^0) = \mathbf{R}\mathbf{Y}^0$. Thus, the initial state is SO(3)-equivariant.

*Inductive Step:* Assume that at step $\tau$, the state is SO(3)-equivariant, i.e., $T_{\mathbf{R}}(\mathbf{Y}^{\tau}) = \mathbf{R}\mathbf{Y}^{\tau}$. From Lemmas 3.1 and 3.2, our model $\mathbf{\Phi}$ is SO(3)-equivariant. Consequently, the predicted target transforms as:

$$T_{\mathbf{R}}(\mathbf{Y}_{\text{pred}}^{\tau}) = \mathbf{\Phi}(T_{\mathbf{R}}(\mathbf{Y}^{\tau}), T_{\mathbf{R}}(\mathbf{X}), \dots) = \mathbf{R}\mathbf{\Phi}(\mathbf{Y}^{\tau}, \mathbf{X}, \dots) = \mathbf{R}\mathbf{Y}_{\text{pred}}^{\tau}. \tag{29}$$

The estimated vector field $\mathbf{u}^{\tau}$ is a linear combination of $\mathbf{Y}_{\text{pred}}^{\tau}$ and $\mathbf{Y}^{\tau}$. Since both terms transform linearly under $T_{\mathbf{R}}$:

$$T_{\mathbf{R}}(\mathbf{u}^{\tau}) = \frac{T_{\mathbf{R}}(\mathbf{Y}_{\text{pred}}^{\tau}) - T_{\mathbf{R}}(\mathbf{Y}^{\tau})}{1 - \tau} = \frac{\mathbf{R}\mathbf{Y}_{\text{pred}}^{\tau} - \mathbf{R}\mathbf{Y}^{\tau}}{1 - \tau} = \mathbf{R}\mathbf{u}^{\tau}. \tag{30}$$

Applying the Euler update rule (Eq. (4)), the state at the next step transforms as:

$$T_{\mathbf{R}}(\mathbf{Y}^{\tau+\Delta\tau}) = T_{\mathbf{R}}(\mathbf{Y}^{\tau}) + T_{\mathbf{R}}(\mathbf{u}^{\tau})\Delta\tau = \mathbf{R}\mathbf{Y}^{\tau} + \mathbf{R}\mathbf{u}^{\tau}\Delta\tau = \mathbf{R}(\mathbf{Y}^{\tau} + \mathbf{u}^{\tau}\Delta\tau) = \mathbf{R}\mathbf{Y}^{\tau+\Delta\tau}. \tag{31}$$

By induction, the generated relative trajectory $\mathbf{Y}^1$ is SO(3)-equivariant.

**Part 2: SE(3)-Equivariance of Absolute Trajectory.** Let $T_g$ denote the global SE(3) transformation $T_g(\mathbf{P}) = \mathbf{R}\mathbf{P} + \mathbf{t}$. The final absolute trajectory is recovered by $\mathbf{F} = \mathbf{Y}^1 + \overline{\mathbf{P}}$. As established in the Preliminaries (Eq. (17)), under $T_g$, the centroid transforms as $T_g(\overline{\mathbf{P}}) = \mathbf{R}\overline{\mathbf{P}} + \mathbf{t}$, and the relative coordinates transform as $T_{\mathbf{R}}(\mathbf{X}) = \mathbf{R}\mathbf{X}$. Combining this with the result from Part 1 ($T_{\mathbf{R}}(\mathbf{Y}^1) = \mathbf{R}\mathbf{Y}^1$), we have:

$$\begin{aligned}
T_g(\mathbf{F}) &= T_{\mathbf{R}}(\mathbf{Y}^1) + T_g(\overline{\mathbf{P}}) \\
&= \mathbf{R}\mathbf{Y}^1 + (\mathbf{R}\overline{\mathbf{P}} + \mathbf{t}) \\
&= \mathbf{R}(\mathbf{Y}^1 + \overline{\mathbf{P}}) + \mathbf{t} \\
&= \mathbf{R}\mathbf{F} + \mathbf{t}.
\end{aligned} \tag{32}$$

This confirms that the final predicted trajectory $\mathbf{F}$ is SE(3)-equivariant with respect to the input $\mathbf{P}$. $\square$

## B. Algorithm

In this section, we provide a structured summary of the proposed GSE-Flow method. To facilitate reproducibility and a clearer understanding of the workflow, we detail the complete algorithmic procedures. Algorithm 1 outlines the training process, including the flow matching setup and loss computation, while Algorithm 2 presents the inference (sampling)

procedure for trajectory prediction. To further facilitate reproducibility, we provide additional details regarding dataset specifications and experimental hyperparameters in the subsequent sections. Furthermore, to support the research community, our source code and the pre-processed datasets will be made publicly available.

---

**Algorithm 1** GSE-Flow Training

---

**Require:** Training dataset $\mathcal{D}$, learning rate $\eta$, Model $\mathbf{\Phi}$ (comprising HCE, FSE, TG).
 1: Initialize parameters $\theta$ of network $\mathbf{\Phi}$.
 2: **repeat**
 3:     **Data Sampling:**
 4:         Sample batch $(\mathbf{P}, \mathbf{H}, \mathbf{E}, \mathbf{F}) \sim \mathcal{D}$.
 5:         Compute history spatiotemporal center $\overline{\mathbf{P}}$ from $\mathbf{P}$.
 6:         Compute relative history $\mathbf{X} \leftarrow \mathbf{P} - \overline{\mathbf{P}}$ and target $\mathbf{Y}^1 \leftarrow \mathbf{F} - \overline{\mathbf{P}}$.
 7:     **Flow Matching Setup:**
 8:         Sample flow time $\tau \sim \mathcal{U}[0, 1]$.
 9:         Sample Gaussian noise $\mathbf{Y}^0 \sim \mathcal{N}(\mathbf{0}, \mathbf{I})$.
10:         Compute noisy state interpolation via Eq. (1):
11:             $\mathbf{Y}^\tau \leftarrow (1 - \tau)\mathbf{Y}^0 + \tau\mathbf{Y}^1$
12:     **Model Forward Pass:**
13:         *// 1. History Context Encoder (HCE)*
14:         $\mathbf{C}_X, \mathbf{C}_H \leftarrow \text{HCE}(\mathbf{X}, \mathbf{H}, \mathbf{E})$
15:         *// 2. Flow State Encoder (FSE)*
16:         $\hat{\mathbf{Y}}^\tau, \hat{\mathbf{Z}}^\tau \leftarrow \text{FSE}(\mathbf{Y}^\tau, \tau, \mathbf{C}_H, \mathbf{X}, \mathbf{E})$
17:         *// 3. Trajectory Generation (TG)*
18:         $\mathbf{Y}^\tau_{\text{pred}} \leftarrow \text{TG}(\mathbf{C}_X, \mathbf{C}_H, \hat{\mathbf{Y}}^\tau, \hat{\mathbf{Z}}^\tau)$
19:     **Optimization:**
20:         Compute loss via Eq. (2):
21:             $\mathcal{L} = ||\mathbf{Y}^1 - \mathbf{Y}^\tau_{\text{pred}}||^2_2$
22:         Update parameters: $\theta \leftarrow \theta - \eta\nabla_\theta\mathcal{L}$
23: **until** converged

---

---

**Algorithm 2** GSE-Flow Sampling (Inference)

---

**Require:** History observation $(\mathbf{P}, \mathbf{H}, \mathbf{E})$, Trained Model $\mathbf{\Phi}$, Number of steps $K$.
 1: **Preprocessing:**
 2:     Compute history spatiotemporal center $\overline{\mathbf{P}}$ from $\mathbf{P}$.
 3:     Compute relative history $\mathbf{X} \leftarrow \mathbf{P} - \overline{\mathbf{P}}$.
 4: **Initialization:**
 5:     Define time schedule $\{\tau_i\}_{i=0}^K$ where $\tau_0 = 0, \tau_K = 1$.
 6:     Sample initial noise $\mathbf{Y}^{\tau_0} \sim \mathcal{N}(\mathbf{0}, \mathbf{I})$.
 7: **for** $i = 0$ to $K - 1$ **do**
 8:     Current time $\tau \leftarrow \tau_i$;   Step size $\Delta\tau \leftarrow \tau_{i+1} - \tau_i$.
 9:     **Predict Clean Target:**
10:         $\mathbf{Y}^\tau_{\text{pred}} \leftarrow \mathbf{\Phi}(\mathbf{Y}^\tau, \mathbf{X}, \mathbf{H}, \mathbf{E}, \tau)$         ▷ Forward pass (HCE,FES,TG)
11:     **Compute Vector Field:**
12:         Estimate velocity $\mathbf{u}^\tau$ via Eq. (3): $\mathbf{u}^\tau \leftarrow \frac{\mathbf{Y}^\tau_{\text{pred}} - \mathbf{Y}^\tau}{1 - \tau}$
13:     **Update State (Euler Step):**
14:         $\mathbf{Y}^{\tau_{i+1}} \leftarrow \mathbf{Y}^\tau + \Delta\tau\mathbf{u}^\tau$
15: **end for**
16: **Final Output:**
17:     Recover absolute trajectory: $\mathbf{F} \leftarrow \mathbf{Y}^{\tau_K} + \overline{\mathbf{P}}$.
18:     **return** Predicted trajectory $\mathbf{F}$.

---

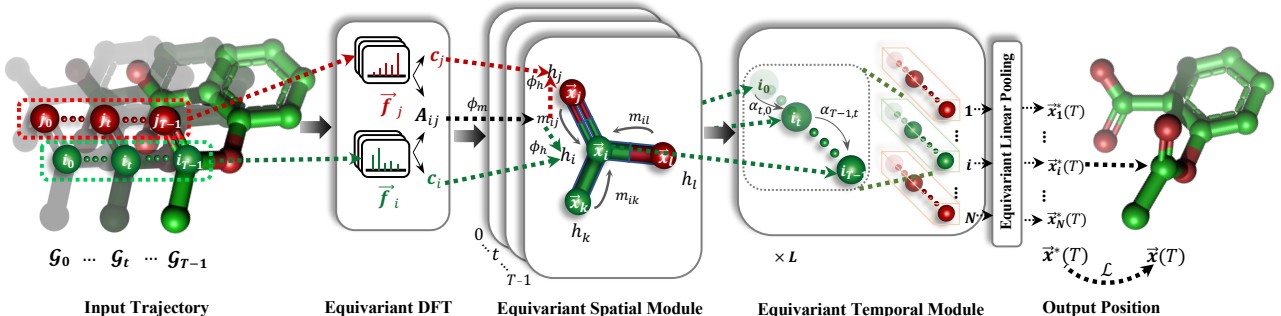

*Figure 4.* Pipeline of ESTAG (Wu et al., 2023) (reproduced for explanation). Our HCE uses the same EDFT–ESM–ETM backbone but removes the final Equivariant Linear Pooling to keep sequence-to-sequence history representations.

## C. Datasets and Experimental Details

To ensure a rigorous and fair comparison, we strictly adhere to the standardized data partitioning protocols established in prior studies (Wu et al., 2023; Yu & Sun, 2024; Huang et al., 2025). The underlying principle governing our configuration is to initially allocate 10%, 5%, and 5% of the total trajectory frames for training, validation, and testing, respectively, subject to specific maximum sequence caps. Under this unified regime, for molecular dynamics tasks—ranging from small molecules in the MD17 dataset to large macromolecules in the MD22 dataset—we apply a consistent truncation rule, limiting the splits to a maximum of 500, 2,000, and 2,000 sequences for training, validation, and testing. Similarly, for the human motion data in CMU MoCap, the subsets are capped at upper limits of 3,000, 600, and 600 sequences, respectively.

Regarding implementation, GSE-Flow is trained using the Adam optimizer with a fixed batch size of 100 across all experiments. The learning rates are tuned specifically for each dataset, set at $5 \times 10^{-3}$ for MD17, MD22 and CMU MoCap. In terms of model architecture, the hidden node feature dimension is fixed at 16. The depth of the equivariant backbones ($\varphi$ and $\psi$ used in HCE and FSE modules) is set to 2 layers for the molecular datasets (MD17 and MD22) and increased to 4 layers for the CMU MoCap dataset to handle its higher complexity. During inference, the model generates trajectories using 10 sampling steps. All of our experiments were conducted on one NVIDIA RTX 4090 GPU.

Following the long-term forecasting protocol in (Xu et al., 2023), we evaluate all methods by predicting $T_f$ future frames given $T_h$ historical frames. To ensure a fair comparison and extend the evaluation to generic SE(3)-equivariant models not originally tailored for this specific long-term forecasting task, we apply lightweight task-alignment adaptations to fit the current protocol. Specifically, for single-step predictors (e.g., TFN, SE(3)-TR, EGNN, FAGNN, EGHN, PT-EvNet, FMN), we augment the inputs with discrete time embeddings and process them via a frame-shared backbone to learn frame-wise representations, which are subsequently mapped to $T_f$ outputs via an equivariant temporal projection head (adopting the design from ESTAG). For models that originally aggregate history into a single output (e.g., ESTAG), we expand their final projection layer to regress the full $T_f$-frame sequence. For architectures requiring a single-frame input (e.g., EGNO), we introduce an equivariant linear layer to compress the $T_h$ historical frames into a unified representation, thereby satisfying their input requirements. Finally, for native multi-frame architectures (e.g., EqMotion, GeoTDM), the results are derived by directly running their official source code without any modification. All baseline evaluations were performed within the resource constraints of four NVIDIA RTX 4090 GPUs.

## D. Architecture Details of $\varphi$ / $\psi$ in HCE / FSE Module

In the HCE and FSE modules, we employ SE(3)-equivariant backbones, denoted as $\varphi$ and $\psi$, to encode the sequence data into geometric representations and node hidden features. The architecture of these backbones is adapted from ESTAG (Wu et al., 2023) and comprises three key components: the Equivariant Discrete Fourier Transform (EDFT) block, the Equivariant Spatial Module (ESM), and the Equivariant Temporal Module (ETM). These blocks are designed to extract high-level representations in the frequency, spatial, and temporal domains, respectively, as illustrated in the pipeline shown in Figure 4 (reproduced from (Wu et al., 2023)). Distinct from the original ESTAG architecture, we remove the final *Equivariant Linear Pooling* head in our backbones $\varphi$ and $\psi$ to retain sequence-to-sequence outputs. We detail these components below.

**Equivariant Discrete Fourier Transform (EDFT).** The EDFT module encodes node-wise temporal dynamics in the

frequency domain while strictly preserving SE(3) symmetry. Let $\mathbf{x}_i(t) \in \mathbb{R}^3$ denote the coordinates of node $i$ at time step $t \in \{1, \ldots, T_h\}$, and let $\bar{\mathbf{x}}(t) = \frac{1}{N} \sum_{j=1}^{N} \mathbf{x}_j(t)$ represent the frame-wise centroid. To ensure translation invariance, the EDFT first centers the trajectory data, followed by a temporal Fourier transform applied to the centered coordinates:

$$\mathbf{f}_i(k) = \sum_{t=1}^{T_h} \exp\left(-\mathrm{i}\frac{2\pi kt}{T_h}\right)(\mathbf{x}_i(t) - \bar{\mathbf{x}}(t)), \quad k = 1, \ldots, T_h, \tag{33}$$

where $\mathrm{i} = \sqrt{-1}$ and $\mathbf{f}_i(k) \in \mathbb{C}^3$ represents the spectral coefficients. Utilizing these coefficients, the module computes SE(3)-invariant frequency statistics, i.e., the frequency-wise cross-correlation $A_{ij}(k)$ and the spectral amplitude $c_i(k)$ as:

$$A_{ij}(k) = \omega_k(\mathbf{h}_i)\omega_k(\mathbf{h}_j)\left|\langle \mathbf{f}_i(k), \mathbf{f}_j(k)\rangle\right|, \qquad c_i(k) = \omega_k(\mathbf{h}_i)\|\mathbf{f}_i(k)\|_2^2, \tag{34}$$

where $\omega_k(\cdot)$ denotes a learnable spectral filter conditioned on the node embeddings $\mathbf{h}_i$ and $\mathbf{h}_j$. These SE(3)-invariant representations serve as inputs for subsequent ESM block.

**Equivariant Spatial Module (ESM).** ESM performs SE(3)-equivariant message passing on each frame $t$ to capture instantaneous spatial dependencies. At layer $\ell$, it computes messages using invariant inputs (node features, squared distances, and EDFT cross-correlation) and updates node features and coordinates in an EGNN-style equivariant manner:

$$\mathbf{m}_{ij} = \phi_m\Big(\mathbf{h}_i^{(\ell)}(t), \mathbf{h}_j^{(\ell)}(t), \|\mathbf{x}_{ij}^{(\ell)}(t)\|_2^2, A_{ij}\Big), \tag{35}$$

$$\mathbf{h}_i^{(\ell+1)}(t) = \mathbf{h}_i^{(\ell)}(t) + \phi_h\Big(\mathbf{h}_i^{(\ell)}(t), \mathbf{c}_i, \sum_{j \neq i} \mathbf{m}_{ij}\Big), \tag{36}$$

$$\mathbf{x}_i^{(\ell+1)}(t) = \mathbf{x}_i^{(\ell)}(t) + \frac{1}{|\mathcal{N}(i)|} \sum_{j \in \mathcal{N}(i)} \mathbf{x}_{ij}^{(\ell)}(t)\, \phi_x(\mathbf{m}_{ij}), \tag{37}$$

where $\mathbf{x}_{ij}^{(\ell)}(t) = \mathbf{x}_i^{(\ell)}(t) - \mathbf{x}_j^{(\ell)}(t)$ and $\phi_m, \phi_h, \phi_x$ are MLPs, $\mathcal{N}(i)$ denotes the neighbors of node $i$. Here $\mathbf{c}_i$ stacks $\{c_i(k)\}_{k=1}^{T_h}$. This design preserves SE(3)-equivariance because coordinate updates are linear combinations of relative displacements with invariant scalar weights.

**Equivariant Temporal Module (ETM).** ETM models temporal dependencies for each node via a causal (forward) attention mechanism. At layer $\ell$, define queries/keys/values from invariant features:

$$\mathbf{q}_i^{(\ell)}(t) = \phi_q(\mathbf{h}_i^{(\ell)}(t)), \quad \mathbf{k}_i^{(\ell)}(s) = \phi_k(\mathbf{h}_i^{(\ell)}(s)), \quad \mathbf{v}_i^{(\ell)}(s) = \phi_v(\mathbf{h}_i^{(\ell)}(s)), \tag{38}$$

and compute masked attention weights for $s \leq t$:

$$\alpha_i^{(\ell)}(t, s) = \frac{\exp\Big(\mathbf{q}_i^{(\ell)}(t)^\top \mathbf{k}_i^{(\ell)}(s)\Big)}{\sum_{s=1}^{t} \exp\Big(\mathbf{q}_i^{(\ell)}(t)^\top \mathbf{k}_i^{(\ell)}(s)\Big)}. \tag{39}$$

The feature and coordinate are then updated by:

$$\mathbf{h}_i^{(\ell+1)}(t) = \mathbf{h}_i^{(\ell)}(t) + \sum_{s=1}^{t} \alpha_i^{(\ell)}(t, s)\, \mathbf{v}_i^{(\ell)}(s), \tag{40}$$

$$\mathbf{x}_i^{(\ell+1)}(t) = \mathbf{x}_i^{(\ell)}(t) + \sum_{s=1}^{t} \alpha_i^{(\ell)}(t, s)\left(\mathbf{x}_i^{(\ell)}(t) - \mathbf{x}_i^{(\ell)}(s)\right) \phi_x(\mathbf{v}_i^{(\ell)}(s)), \tag{41}$$

where $d$ is the query/key dimension and $\phi_q, \phi_k, \phi_v, \phi_x$ are MLPs. These operations maintain the SE(3)-equivariance since they aggregates relative temporal displacements with invariant scalar weights. In the original ESTAG, the final head pools the temporal dimension to output a single frame; in our model, we remove that pooling and keep the per-time-step outputs as the encoded history context.

# E. More Experimental Results on Generality

To evaluate the generality of GSE-Flow, we integrate it with various deterministic baselines (e.g., EGNN, EGHN, EGNO, ESTAG) by employing them as the sequence encoders $\varphi$ (Eq. 5) and $\psi$ (Eq. 6). The consistent gains observed on the MD17 dataset, as shown in Figure 3, demonstrate that augmenting deterministic backbones with our probabilistic flow matching framework significantly enhances their predictive capabilities. We provide the complete numerical results for MD17 in Table 5. Furthermore, we extend this evaluation to the MD22 and CMU MoCap datasets, detailing ADE and FDE scores in Tables 7–6. In these results, 'Base' denotes the original deterministic models, while 'GSE-Base' refers to the corresponding models integrated with GSE-Flow. The consistent performance improvements across these diverse architectures validate the generality and effectiveness of our approach.

*Table 5.* Quantitative comparison on the MD17 dataset across different backbone architectures. We evaluate the impact of integrating GSE-Flow (denoted as 'GSE-Base') with deterministic baselines (denoted as 'Base'). Results are reported as ADE/FDE ($\times 10^{-2}$).

| Dataset | Setting | EGNN | EGNO | EGHN | ESTAG |
|---|---|---|---|---|---|
| Aspirin | Base | 10.13/15.12 | 7.27/12.18 | 9.11/13.83 | 6.77/10.87 |
| | GSE-Base | 7.80/12.26 | 6.07/10.47 | 7.09/12.22 | **4.85/7.96** |
| Benzene | Base | 4.81/6.68 | 3.35/5.21 | 3.32/5.64 | 3.40/5.95 |
| | GSE-Base | 3.73/5.58 | 1.60/2.92 | 1.57/2.80 | **1.38/2.50** |
| Ethanol | Base | 5.16/7.10 | 5.11/7.11 | 5.05/7.13 | 4.83/6.88 |
| | GSE-Base | 4.97/7.13 | 4.64/6.71 | 4.97/7.09 | **4.48/6.60** |
| Malonaldehyde | Base | 7.09/10.88 | 6.64/10.55 | 7.36/11.36 | 6.53/10.09 |
| | GSE-Base | 6.10/9.43 | 5.77/9.34 | 5.92/9.98 | **5.68/9.17** |
| Naphthalene | Base | 7.37/9.65 | 6.14/8.76 | 6.54/8.13 | 6.09/7.97 |
| | GSE-Base | 4.81/5.92 | 3.45/4.94 | 3.82/5.29 | **3.12/4.34** |
| Salicylic | Base | 9.77/13.98 | 6.62/11.18 | 9.79/12.89 | 8.08/12.98 |
| | GSE-Base | 5.71/8.75 | 4.64/7.10 | 4.73/6.55 | **4.09/6.01** |
| Toluene | Base | 5.84/7.05 | 5.59/7.61 | 5.72/6.95 | 5.49/7.41 |
| | GSE-Base | 4.18/5.12 | 3.44/4.77 | 3.57/4.62 | **3.09/4.17** |
| Uracil | Base | 5.61/7.34 | 5.09/7.48 | 6.02/7.06 | 4.83/6.22 |
| | GSE-Base | 4.45/5.69 | 3.87/5.02 | 3.99/5.24 | **3.43/4.61** |

*Table 6.* Quantitative comparison on CMU MoCap dataset across different backbone architectures. We evaluate the impact of integrating GSE-Flow (denoted as 'GSE-Base') with deterministic baselines (denoted as 'Base'). Results are reported as ADE/FDE ($\times 10^{-1}$).

| Dataset | Setting | EGNN | EGNO | EGHN | ESTAG |
|---|---|---|---|---|---|
| Basketball | Base | 99.20/154.52 | 38.98/73.81 | 39.94/68.06 | 38.85/67.49 |
| | GSE-Base | 82.30/115.13 | 34.78/66.64 | 37.38/61.86 | **32.39/57.63** |
| Walk | Base | 7.22/10.80 | 3.63/6.06 | 6.25/9.11 | 3.27/5.20 |
| | GSE-Base | 11.92/18.15 | 2.71/4.36 | **2.57/4.18** | 2.59/4.25 |

# F. Runtime and GPU Memory

We benchmark training/inference footprints against competitive methods on MD17 *Aspirin & AT-AT* and CMU MoCap *Walk*; entries follow the format Asp./AT-AT/Walk. Table 8 reports training & inference memory (MB), wall-clock milliseconds per optimizer/inference iteration, and paired ADE/FDE (same conventions as Tables 5–6). The continuous-time rollout of GSE-Flow therefore incurs higher inference latency than deterministic single-shot predictors because iterative integration substitutes for a closed-form one-pass forward map. Nevertheless, aided by deliberately lightweight ESTAG-derived equivariant backbones, GSE-Flow preserves a comparatively lean memory budget and achieves accuracy highly competitive or superior to heavyweight baselines (*e.g.,* FAGNN, FMN) while narrowing the inference memory gap materially. Across the regimes shown the pipeline therefore remains broadly compatible with low-latency deployment requirements despite its generative design.

*Table 7.* Quantitative comparison on the MD22 dataset across different backbone architectures. We evaluate the impact of integrating GSE-Flow (denoted as 'GSE-Base') with deterministic baselines (denoted as 'Base'). Results are reported as ADE/FDE ($\times 10^{-2}$).

| Dataset | Setting | EGNN | EGNO | EGHN | ESTAG |
|---|---|---|---|---|---|
| AT-AT | Base | 21.73/33.46 | 13.58/23.51 | 15.67/25.10 | 14.05/23.10 |
| | GSE-Base | 17.88/30.69 | 11.68/21.46 | 12.21/21.35 | **9.15/16.32** |
| AT-AT-CG-CG | Base | 22.47/37.48 | 14.46/25.30 | 16.40/27.44 | 15.01/25.19 |
| | GSE-Base | 21.15/36.48 | 12.75/23.15 | 13.13/23.09 | **10.35/19.09** |
| Ac-Ala$_3$-NHMe | Base | 21.65/34.89 | 15.77/26.79 | 16.87/26.58 | 14.24/23.11 |
| | GSE-Base | 18.27/29.94 | 12.94/22.23 | 14.24/24.29 | **10.94/19.01** |
| DHA | Base | 20.45/33.23 | 15.67/27.30 | 16.64/27.60 | 15.45/25.99 |
| | GSE-Base | 18.64/31.78 | 13.16/23.21 | 15.24/27.71 | **11.45/20.86** |
| Buckyball Catcher | Base | 16.07/23.96 | 12.21/18.54 | 13.02/18.35 | 11.24/15.91 |
| | GSE-Base | 14.24/22.46 | 10.34/15.50 | 10.30/14.03 | **7.52/12.20** |
| Double-walled Nanotube | Base | 19.65/28.14 | 15.50/21.87 | 16.91/22.07 | 13.67/19.21 |
| | GSE-Base | 14.39/19.50 | 14.31/20.98 | 15.26/22.21 | **10.26/15.70** |
| Stachyose | Base | 20.87/32.30 | 17.71/28.95 | 19.59/30.29 | 17.52/27.61 |
| | GSE-Base | 19.27/30.56 | 16.68/27.83 | 16.65/27.69 | **14.66/24.39** |

*Table 8.* Efficiency & accuracy benchmarks. Train/Infer Memory (MB) and Time (ms/iter), ADE/FDE ($\times 10^{-2}$ for molecular entries, $\times 10^{-1}$ for Walk) with Asp./AT-AT/Walk triples.

| Method | Train Mem. Asp./AT./Walk | Infer Mem. Asp./AT./Walk | Train Time Asp./AT./Walk | Infer Time Asp./AT./Walk | ADE Asp./AT./Walk | FDE Asp./AT./Walk |
|---|---|---|---|---|---|---|
| EGNN | 92.6/252.0/333.2 | 44.1/100.9/72.0 | 9.9/12.2/21.3 | 3.3/3.1/5.2 | 10.13/21.73/7.22 | 15.12/33.46/10.80 |
| ESTAG | 142.5/402.4/558.5 | 46.5/111.9/81.8 | 26.5/13.7/43.9 | 5.9/4.2/7.7 | 6.77/14.05/3.27 | 10.87/23.10/5.20 |
| EqMotion | 151.7/1115.4/1282.9 | 44.1/231.5/162.4 | 32.8/27.3/54.4 | 7.8/9.8/13.7 | 7.05/11.59/5.85 | 10.62/18.37/7.52 |
| EGNO | 594.5/2957.2/1154.4 | 218.3/1066.0/223.9 | 24.6/78.1/51.9 | 4.5/17.9/7.6 | 7.27/13.58/3.63 | 12.18/23.51/6.06 |
| EGHN | 1194.4/6186.3/2432.0 | 263.9/1317.6/273.9 | 48.0/211.2/95.4 | 12.9/61.2/25.4 | 9.11/15.67/6.25 | 13.83/25.10/9.11 |
| FMN | 3165.3/15515.2/3448.4 | 1365.8/7033.9/1491.6 | 57.0/201.4/58.9 | 12.9/50.8/14.2 | 8.32/14.61/3.03 | 11.72/21.74/4.88 |
| FAGNN | 3261.3/17235.9/6158.8 | 1561.2/8248.2/1653.2 | 40.3/227.6/82.9 | 27.0/160.4/54.0 | 6.92/13.62/3.12 | 9.44/20.90/4.42 |
| Ours | 473.4/1384.8/1571.2 | 94.4/254.4/162.7 | 48.8/74.4/131.3 | 53.1/59.2/125.5 | **4.85/9.15/2.59** | **7.96/16.32/4.25** |

