# OpenReview forum: "Flow for Future: Geometric SE(3)-Equivariant Flow Matching for 3D Trajectory Prediction"
_ICML.cc/2026/Conference — ICML 2026 regular_

### Official Review · Reviewer_HRKe · 2026-02-28

**Soundness:** 3
**Presentation:** 3
**Significance:** 3
**Originality:** 2
**Overall Recommendation:** 4
**Confidence:** 4

**Summary:**

The authors propose GSE-Flow, a generative framework that combines conditional flow matching with SE(3)-equivariant networks for 3D trajectory prediction. The method addresses the challenge of conditioning continuous generative flows on discrete historical observations by employing an equivariant backbone to extract sequence features, modulating the flow states with time embeddings, and fusing the historical context with the evolving flow via a tensorization-based attention mechanism. The paper demonstrates strong empirical performance across molecular and human motion datasets.

**Compliance With Llm Reviewing Policy:**

Affirmed.

**Final Justification:**

The authors has addressed my concerns, I recommend this paper to be accepted considering the rebuttal and their contributions.

**Key Questions For Authors:**

- Regarding the lightweight adaptations made to the deterministic baselines (e.g., EGNN, ESTAG) for long-term forecasting, could you provide more details on the hyperparameter tuning process to ensure a strictly fair comparison and rule out any performance gaps caused by asymmetrical tuning?
- The "Geometry-Feature Tensorization" module appears to utilize an outer product followed by dot-product attention. Could you clarify if there is a fundamental mathematical distinction between this module and standard cross-attention mechanisms operating on concatenated geometric and semantic features?

**Limitations:**

The authors have adequately discussed standard experimental limitations. However, it would be beneficial to include a brief discussion on the potential risks of error accumulation during the numerical integration steps over longer prediction horizons, as well as any expressivity bottlenecks inherited directly from the chosen equivariant graph neural network backbone

**Strengths And Weaknesses:**

The proposed architecture is technically sound, and the experiments demonstrate solid empirical gains across multiple datasets of varying scales. However, a minor weakness lies in the evaluation of deterministic baselines, which were adapted for multi-step forecasting via lightweight modifications; it would be helpful to clarify if these baselines received the same level of rigorous hyperparameter tuning as the proposed method to ensure a strictly fair comparison. Addressing the uncertainty of geometric trajectory prediction via generative modeling is a relevant and practically valuable direction. While the algorithmic components are largely integrations of existing equivariant network modules and standard flow matching objectives, the combined system provides a robust empirical baseline that future research can build upon. The mathematical guarantees for equivariance are largely inherited from the chosen backbone, leaving an opportunity for future theoretical work to explicitly define the optimal transport mapping between historical discrete contexts and continuous future manifolds.

---

> ### Author Rebuttal · Authors · 2026-03-30
>
> **Q1.Hyperparameter tuning & fair comparison.**
>
> While structural details of lightweight adaptations for multi-frame forecasting are provided in Appendix C, we clarify the tuning process here.
>
> To ensure fair comparison, all models were implemented based on official codebases with batch size 100. For natively multi-frame baselines (EqMotion & GeoTDM), we used their highly optimized default hyperparameters. For adapted deterministic baselines, we observed stable convergence over range $10^{-4}$ to $5 \times 10^{-3}$, and adopted learning rate $5\times10^{-3}$ for their methods and ours, avoiding asymmetric learning-rate tuning.
>
> For network depth, we retained the original layer counts from their official single-frame designs, as increasing depth in these GNN baselines tended to worsen over-smoothing and, in some settings, caused OOM. To rule out the possibility that the adapted baselines were bottlenecked by limited network capacity, we conducted capacity-scaling experiment by varying feature dimension $d$. Due to space limits, we report ADE & FDE on Aspirin ($\times10^{-2}$) & Walk ($\times10^{-1}$) below, formatted as values for $d=16/32/64/128$. As shown, $d=16$ yields the best stability-performance trade-off. Even when baselines peak at higher capacities, they remain inferior to ours at $d=16$. Thus, setting $d=16$ uniformly provides a consistent and practically fair capacity setting.
>
> ||ADE(Aspirin)|FDE(Aspirin)|ADE(Walk)|FDE(Walk)|
> |---|---|---|---|---|
> |FA-GNN|6.92/6.10/7.24\/7.74|9.44\/8.13\/9.83\/10.35|3.12\/2.79\/3.05\/3.28|4.42\/4.42\/4.35\/4.86|
> |FMN|8.32\/7.46\/7.63\/8.31|11.72\/10.67\/11.07\/11.79|3.03\/2.78\/OOM\/OOM|4.88\/4.34\/OOM\/OOM|
> |ESTAG|6.77\/8.28\/10.80\/12.11|10.87\/13.27\/16.09\/17.92|3.27\/3.06\/5.53\/71.70|5.20\/4.87\/9.12\/62.94|
> |EGNN|10.13\/10.13\/10.83\/12.94|15.12\/15.00\/16.29\/18.50|7.22\/7.69\/6.92\/8.26|10.80\/11.30\/11.09\/12.23|
> |EGHN|9.11\/10.28\/17.98\/41.33|13.83\/15.69\/19.79\/84.21|6.25\/16.84\/128.00\/225.99|9.11\/19.52\/146.78\/208.37|
> |EGNO|7.27\/10.00\/10.32\/12.26|12.18\/15.75\/15.49\/17.95|3.63\/4.22\/3.68\/152.33|6.06\/6.75\/5.93\/156.69|
> |Ours|**4.85**\/5.74\/6.26\/7.04|**7.96**\/9.55\/10.94\/12.13|2.59\/2.40\/**2.35**\/2.40|4.25\/3.91\/3.88\/**3.86**|
>
> *Note: Values formatted for $d=16/32/64/128$.*
>
> **Q2.Mathematical distinction of GFT vs standard cross-attention on concatenated features.**
>
> Let $\mathbf{C}_X,\mathbf{C}_H$ denote geometric and semantic history states, and $\mathbf{Y},\mathbf{Z}$ denote those of the flow. Standard cross-attention concatenates features, yielding an additive inner product $\langle [\mathbf{C}_X \parallel \mathbf{C}_H],[\mathbf{Y}\parallel\mathbf{Z}]\rangle=\langle \mathbf{C}_X,\mathbf{Y} \rangle+\langle \mathbf{C}_H, \mathbf{Z}\rangle$. This treats spatial and semantic similarities independently, where strong geometric proximity can erroneously dominate attention score despite a severe semantic mismatch, and vice versa. Furthermore, unconstrained linear projections in standard attention on concatenated features mix type-1 covariant coordinates with type-0 invariant scalars, which does not in general preserve equivariance.
>
> Conversely, GFT couples features via tensor product. In SO(3) equivariant networks, coupling irreducible representations (irreps) of type-1 vectors ($\mathbf{C}_X, \mathbf{Y}$) and type-0 scalars ($\mathbf{C}_H, \mathbf{Z}$) necessitates a tensor product to construct a valid joint space. The inner product in this joint space simplifies to a bilinear form $\langle\text{vec}(\mathbf{C}_X \otimes \mathbf{C}_H), \text{vec}(\mathbf{Y} \otimes \mathbf{Z}) \rangle = \langle \mathbf{C}_X, \mathbf{Y} \rangle \cdot \langle \mathbf{C}_H, \mathbf{Z} \rangle$. Acting as a mutual gate, this bilinear form forces semantics to modulate geometry and vice versa, and high attention needs both to align. Furthermore, this formulation inherently guarantees equivariance.
>
> Empirically, replacing GFT with standard cross-attention (GSE-Flow-CA below) yields inferior results, confirming that this design is insufficient for complex geometric-semantic interactions.
>
> ||ADE/FDE (Asp.,$\times10^{-2}$)|ADE/FDE (AT-AT,$\times10^{-2}$)|ADE/FDE (Walk,$\times10^{-1}$)|
> |:---|:---|:---|:---|
> |GSE-Flow-CA|13.65\/19.54|21.50\/32.84|3.28\/5.38|
> |GSE-Flow|4.85\/7.96|9.15\/16.32|2.59\/4.25|
>
> **Q3. Discussion on error accumulation & bottlenecks of equivariant GNN.**
>
> We will add the limitations. First, continuous flow matching accumulates ODE discretization errors over long horizons and causes trajectory drift, which future work can mitigate by consistency distillation. Second, equivariant GNN backbone restricts expressivity because relying on pairwise distances limits higher order geometric modeling. Furthermore, strict SE(3) equivariance becomes a structural bottleneck when incorporating asymmetric contexts like HD maps & traffic rules that inherently break spatial symmetry.
>
> **Thanks for the suggestions. We will add above analysis in revision.**

---

> > ### Author Rebuttal · Reviewer_HRKe · 2026-04-01
> >
> > Thanks for your response, I will keep my score

---

> > > ### Author Response · Authors · 2026-04-04
> > >
> > > Thank you very much for your positive feedback and for your careful consideration of our rebuttal. We will continue refining the paper and ensure that the relevant clarifications and additional results are properly incorporated into the final version. If you have any further questions, we would be very happy to provide additional details.

---

### Official Review · Reviewer_tzbx · 2026-03-04

**Soundness:** 3
**Presentation:** 3
**Significance:** 3
**Originality:** 3
**Overall Recommendation:** 4
**Confidence:** 2

**Summary:**

This paper proposes GSE-Flow, an SE(3)-equivariant flow matching framework for 3D trajectory prediction. The method integrates a history context encoder, a flow state encoder, and a trajectory generation module to enforce geometric consistency throughout the generative process.

**Compliance With Llm Reviewing Policy:**

Affirmed.

**Final Justification:**

Thanks for authors' effort. I will keep my score.

**Key Questions For Authors:**

1. Could the authors provide an ablation study analyzing the impact of different numbers of sampling steps (e.g., 1, 5, 10, 20) on prediction accuracy?
2. Compared to simple methods that directly concatenate historical features and flow states, how much improvement in prediction accuracy does your proposed GFT achieve?

**Strengths And Weaknesses:**

**Strength**:
1. Novel Integration of Flow Matching with Equivariance: The paper presents a principled combination of flow matching and SE(3)-equivariant networks, which is a timely and important contribution to the field of geometric trajectory prediction.
2. Strong Empirical Performance: GSE-Flow consistently outperforms a wide range of baselines on multiple benchmarks, including MD17, MD22, and CMU MoCap, demonstrating its generalization across domains and scales.
3. Theoretical Rigor: The authors provide formal proofs of equivariance for each module and the entire generative process, ensuring the model's geometric consistency is mathematically grounded.

**Weakness**:
1. The paper lacks an analysis of how the number of sampling steps affects prediction accuracy.
2. The ablation study lacks a direct comparison against the "shallow concatenation" baseline criticized in the paper. I recommend adding this comparison to validate the necessity of deep geometric coupling.

---

> ### Author Rebuttal · Authors · 2026-03-30
>
> **Q1. Ablation study analyzing the impact of sampling steps on prediction accuracy.**
>
> Thanks for the suggestion. To systematically evaluate the impact of sampling steps ($N$) on prediction accuracy, we report the ADE/FDE on the Aspirin, AT-AT, and Walk datasets in the following tables. Additionally, we report the inference computational cost in terms of Time (ms) and Peak GPU Memory (MB).
>
> As shown in the tables, our model does not require a large number of sampling steps to achieve strong performance, though the optimal step count varies across datasets. Specifically, performance on Walk largely saturates after 4-6 steps, peaks at 14 steps for AT-AT, and reaches its best ADE/FDE at steps 18/12 for Aspirin. Furthermore, while inference time scales linearly with $N$, peak memory increases only mildly. Although the per-dataset optimum varies, our default setting ($N=10$) consistently achieves near-best accuracy across all three datasets while incurring substantially lower latency than larger step counts, making it a robust trade-off between accuracy and inference cost. We will include this discussion in the revision.
>
> Table A. Results on Aspirin.
> | Steps ($N$) | ADE ($\times 10^{-2}$) | FDE ($\times 10^{-2}$) | Time (ms) | GPU Memory (MB) |
> | :---: | :---: | :---: | :---: | :---: |
> | 1 | 5.36 | 8.84 | 6.6 | 91.55 |
> | 2 | 4.89 | 7.73 | 11.9 | 92.00 |
> | 4 | 5.01 | 8.33 | 22.3 | 92.59 |
> | 6 | 5.12 | 8.21 | 32.5 | 93.19 |
> | 8 | 5.43 | 8.77 | 42.6 | 94.31 |
> | 10 | 4.85 | 7.96 | 53.1 | 94.40 |
> | 12 | 4.86 | **7.45** | 67.9 | 96.10 |
> | 14 | 5.12 | 8.15 | 73.9 | 96.40 |
> | 16 | 5.01 | 8.10 | 84.6 | 97.42 |
> | 18 | **4.71** | 8.31 | 97.8 | 97.81 |
> | 20 | 4.94 | 7.58 | 105.6 | 97.36 |
>
>
> Table B. Results on AT-AT.
> | Steps ($N$) | ADE ($\times 10^{-2}$) | FDE ($\times 10^{-2}$) | Time (ms) | GPU Memory (MB) |
> | :---: | :---: | :---: | :---: | :---: |
> | 1 | 10.17 | 18.13 | 7.7 | 245.14 |
> | 2 | 12.09 | 21.38 | 13.4 | 246.46 |
> | 4 | 10.53 | 18.51 | 24.8 | 248.33 |
> | 6 | 10.42 | 18.27 | 36.3 | 251.62 |
> | 8 | 9.71 | 16.85 | 47.8 | 252.36 |
> | 10 | 9.15 | 16.32 | 59.2 | 254.41 |
> | 12 | 9.47 | 16.84 | 70.0 | 255.69 |
> | 14 | **8.87** | **15.25** | 82.4 | 258.48 |
> | 16 | 9.50 | 16.55 | 93.2 | 260.62 |
> | 18 | 10.16 | 18.04 | 104.2 | 261.52 |
> | 20 | 9.82 | 17.15 | 115.8 | 263.26 |
>
>
> Table C. Results on Walk.
> | Steps ($N$) | ADE ($\times 10^{-1}$) | FDE ($\times 10^{-1}$) | Time (ms) | GPU Memory (MB) |
> | :---: | :---: | :---: | :---: | :---: |
> | 1 | 2.89 | 4.84 | 18.9 | 156.61 |
> | 2 | 2.71 | 4.39 | 29.9 | 157.69 |
> | 4 | 2.62 | **4.22** | 54.4 | 159.38 |
> | 6 | **2.59** | **4.22** | 81.3 | 160.40 |
> | 8 | 2.73 | 4.36 | 98.4 | 161.90 |
> | 10 | **2.59** | 4.25 | 125.5 | 162.70 |
> | 12 | 2.64 | 4.31 | 157.9 | 166.21 |
> | 14 | 2.64 | 4.29 | 166.2 | 165.89 |
> | 16 | 2.65 | 4.37 | 190.3 | 169.62 |
> | 18 | 2.66 | 4.35 | 215.2 | 168.91 |
> | 20 | 2.69 | 4.37 | 236.4 | 170.89 |
>
> **Q2. Ablation study by shallow concatenation in the trajectory generation (TG) module.**
>
> Thanks for this suggestion. To validate the necessity of our deep geometric coupling, we conduct ablation studies by replacing the TG module with two "shallow concatenation" baselines:
>
> (1) Replacing TG with concatenation followed by equivariant block: We directly concatenate the historical and flow states (including both geometric coordinates and invariant features) along the temporal dimension. The concatenated representation is then fed into a standard equivariant backbone (specifically, the ESTAG block used in our HCE and FSE modules) to predict the future state.
>
> (2) Replacing TG with concatenation followed by cross-attention: We concatenate the historical context and the flow states (including both geometric coordinates and invariant features) along the temporal dimension. The concatenated representation is then fed into a standard cross-attention block for future state prediction, where the flow state is taken as the query, and the historical context serves as the key and value.
>
> The ADE/FDE on Aspirin ($\times 10^{-2}$), AT-AT ($\times 10^{-2}$) and Walk ($\times 10^{-1}$) are detailed in following table, where both shallow concatenation variants suffer performance degradation. This indicates that simple concatenation is insufficient to capture the complex geometric dependencies between historical contexts and flow states. These results empirically validate the necessity and effectiveness of our deep context-flow fusion design. We will include these ablation studies and the discussion in the revised manuscript.
>
> | Variant | ADE (Aspirin) | FDE (Aspirin) | ADE (AT-AT) | FDE (AT-AT) | ADE (Walk) | FDE (Walk)|
> | :--- | :---: | :---: | :---: | :---: | :---: | :---: |
> | (1)  | 7.50 | 12.74 | 12.73 | 21.69 | 4.24 | 6.89 |
> | (2)  | 13.65 | 19.54 | 21.50 | 32.84 | 3.28 | 5.38 |
> | GSE-Flow | **4.85** | **7.96** | **9.15** | **16.32** | **2.59** | **4.25** |

---

> > ### Author Rebuttal · Reviewer_tzbx · 2026-04-02
> >
> > No

---

> > > ### Author Response · Authors · 2026-04-04
> > >
> > > Thank you very much for your positive feedback and for your careful consideration of our rebuttal. We will continue refining the paper and ensure that the relevant clarifications and additional results are properly incorporated into the final version. If you have any further questions, we would be very happy to provide additional details.

---

### Official Review · Reviewer_FnLA · 2026-03-12

**Soundness:** 3
**Presentation:** 3
**Significance:** 3
**Originality:** 2
**Overall Recommendation:** 4
**Confidence:** 4

**Summary:**

This paper proposes GSE-Flow, a conditional flow matching framework for SE(3)-equivariant 3D trajectory prediction. The method reformulates future trajectory forecasting as a continuous generative process in a centered relative coordinate system, and introduces three main components to bridge deterministic history with stochastic flow evolution: a History Context Encoder (HCE), a Flow State Encoder (FSE) with coherent sequence encoding and time-modulated embedding, and a Trajectory Generation (TG) module with geometry-feature tensorization and context-flow fusion. The paper also provides equivariance claims for the main modules and reports strong empirical performance on MD17, MD22, and CMU MoCap, along with ablations and evidence that the framework can enhance deterministic backbones.

**Compliance With Llm Reviewing Policy:**

Affirmed.

**Key Questions For Authors:**

1. The paper is motivated as a generative approach for uncertain future dynamics, yet the evaluation is almost entirely based on ADE/FDE. Could the authors provide multi-sample evaluations that directly assess generative quality, such as diversity, coverage, calibration, or best-of-K performance? If such results show clear benefits over deterministic or alternative generative baselines, my assessment of both soundness and significance would improve.

2. The method is presented as both accurate and stable, but the paper does not provide a systematic efficiency analysis. Could the authors report training cost, inference cost, sampling steps, and memory usage relative to strong baselines? A favorable cost–performance trade-off would strengthen the practical significance of the work; conversely, a very high generation cost would narrow its impact.

3. Boundary Conditions in Temporal Velocity Approximation: In Eq. (7), the instantaneous velocity $\tilde{V}^\tau$ is explicitly derived from the flow coordinate $\tilde{Y}^\tau$ using a finite difference mechanism ($\Delta_t$). How does the model mathematically accommodate boundary temporal conditions (specifically at the precise initialization step $t=1$ and the final forecasting horizon $t=T_f$) to prevent severe numerical instability when the noise schedule dictates $\tau \to 0$?

4. For molecular trajectory prediction in particular, could the authors include additional physically meaningful evaluations, such as stability over longer horizons or structure-preservation diagnostics, beyond displacement error alone? If the method also performs well on such metrics, I would view the contribution as more significant and more convincing scientifically.

**Limitations:**

No. The paper does not adequately discuss its limitations or potential negative societal impact. The current impact statement is overly brief and does not engage with realistic constraints or risks. A stronger discussion should include at least the following points:

1. the method’s computational cost and sampling overhead relative to deterministic predictors;

2. the limited evidence for generative quality beyond ADE/FDE;

3. possible failure modes under distribution shift, long-horizon rollout, or highly multimodal futures.

**Strengths And Weaknesses:**

Strengths: The technical formulation is coherent and internally well motivated. The paper clearly states the symmetry-preserving objective, rewrites the task in a centered relative coordinate system, formulates trajectory prediction through conditional flow matching, and provides a modular design whose roles are conceptually distinct. The manuscript also includes explicit equivariance statements for the time-modulated state, the attention mechanism, and the full generation process. On the empirical side, the evaluation spans molecular and human-motion benchmarks, and the ablation study is reasonably comprehensive, covering the major design choices.

Weaknesses: The empirical validation is strong in terms of point-wise displacement errors, but it does not yet fully validate the generative motivation of the method. The paper motivates the framework partly by the indeterminacy and distributional nature of future motion, yet the reported evaluation is limited to ADE/FDE; there is no explicit assessment of sample diversity, calibration, multimodal coverage, or likelihood-related behavior. In addition, the paper claims stability and practical advantages, but it does not provide a systematic runtime, memory, or sampling-cost comparison. Finally, while the paper states that detailed proofs and algorithms are provided in the appendix, reproducibility would be stronger if implementation-sensitive details were surfaced more prominently in the main paper.

---

> ### Author Rebuttal · Authors · 2026-03-30
>
> **Q1.Multi-sample evaluation on generative quality.**
>
> In response to Reviewer gjRG Q3, we report  minADE$_K$, minFDE$_K$ and APD to evaluate best-of-K performance, coverage proxy, and diversity, respectively, while Q4 of this response further reports NLL and Bond Length RMSE for likelihood-related behavior and physical plausibility.
>
> **Q2.Runtime, memory, sampling-cost.**
>
> Please see response to Reviewer tzbx Q1  for sampling-cost with different steps. We report runtime (ms\/iter), GPU memory (MB), and  ADE/FDE results on Aspirin & AT-AT ($\times 10^{-2}$ ) and  Walk ($\times 10^{-1}$) against competitive baselines as below.
>
> |Method|Train Mem.(MB) (Asp./AT-AT/Walk)|Infer Mem.(MB) (Asp./AT-AT/Walk)|Train Time(ms) (Asp./AT-AT/Walk)|Infer Time(ms) (Asp./AT-AT/Walk)|ADE (Asp./AT-AT/Walk)|FDE (Asp./AT-AT/Walk)|
> |:---|:---|:---|:---|:---|:---|:---|
> |EGNN|92.6\/252.0\/333.2|44.1\/100.9\/72.0|9.9\/12.2\/21.3|3.3\/3.1\/5.2|10.13\/21.73\/7.22|15.12\/33.46\/10.80|
> |ESTAG|142.5\/402.4\/558.5|46.5\/111.9\/81.8|26.5\/13.7\/43.9|5.9\/4.2\/7.7|6.77\/14.05\/3.27|10.87\/23.10\/5.20|
> |EqMotion|151.7\/1115.4\/1282.9|44.1\/231.5\/162.4|32.8\/27.3\/54.4|7.8\/9.8\/13.7|7.05\/11.59\/5.85|10.62\/18.37\/7.52|
> |EGNO|594.5\/2957.2\/1154.4|218.3\/1066.0\/223.9|24.6\/78.1\/51.9|4.5\/17.9\/7.6|7.27\/13.58\/3.63|12.18\/23.51\/6.06|
> |EGHN|1194.4\/6186.3\/2432.0|263.9\/1317.6\/273.9|48.0\/211.2\/95.4|12.9\/61.2\/25.4|9.11\/15.67\/6.25|13.83\/25.10\/9.11|
> |FMN|3165.3\/15515.2\/3448.4|1365.8\/7033.9\/1491.6|57.0\/201.4\/58.9|12.9\/50.8\/14.2|8.32\/14.61\/3.03|11.72\/21.74\/4.88|
> |FAGNN|3261.3\/17235.9\/6158.8|1561.2\/8248.2\/1653.2|40.3\/227.6\/82.9|27.0\/160.4\/54.0|6.92\/13.62\/3.12|9.44\/20.90\/4.42|
> |Ours|473.4\/1384.8\/1571.2|94.4\/254.4\/162.7|48.8\/74.4\/131.3|53.1\/59.2\/125.5|**4.85\/9.15\/2.59**|**7.96\/16.32\/4.25**|
>
>
> Although the continuous-time formulation requires iterative ODE integration and thus incurs higher inference latency than single-step deterministic baselines, GSE-Flow maintains a highly efficient memory footprint. Compared to other high-accuracy models like FAGNN and FMN, our method significantly reduces inference memory while delivering superior performance.
>
> **Q3.Boundary conditions & numerical stability of velocity in Eq.(7).**
>
> We ensure numerical stability by explicitly addressing the boundaries across both the sequence temporal dimension ($t$) and the flow time dimension ($\tau$):
>
> For the sequence temporal boundaries, to prevent ill-posed queries when computing finite differences ($\Delta_t$) along the sequence, the velocity is explicitly initialized to zero at the precise initialization step ($t=1$) and computed via strictly backward differences at the final forecasting horizon ($t=T_f$). This guarantees valid velocity computation at every frame.
>
> For the flow time boundary ($\tau \to 0$), the formulation intrinsically precludes numerical explosion. The input state reduces to the standard Gaussian prior $\mathcal{N}(\mathbf{0}, \mathbf{I})$, which the equivariant backbone $\psi$ (acting as a continuous mapping) processes into bounded flow coordinates $\tilde{\mathbf{Y}}^\tau$. Since the forward pass consists solely of continuous operations and strictly excludes $\tau$-parameterized singularities (e.g., division by $\tau$), applying the linear finite difference operator to the bounded $\tilde{\mathbf{Y}}^\tau$ guarantees a strictly finite velocity.
>
> Furthermore, the scalar $\beta_{eq}$ in Eq. (7) is learned as a continuous function of $\tau$. Acting as an adaptive gate, it enables the network to implicitly modulate the kinematic contributions across the flow trajectory in a purely data-driven manner.
>
> **Q4.Physically meaningful evaluations.**
>
> We additionally report NLL and Bond Length RMSE ($\times 10^{-2}$) to assess distributional quality and structure preservation. The results show competitive likelihood estimates and strong structural consistency, providing complementary evidence beyond displacement error.
>
> |Method|NLL↓(Asp./AT-AT/Walk)|Bond Length RMSE↓(Asp./AT-AT/Walk)|
> |:---|:---:|:---:|
> |EGHN|2.7620/6.7885/2.7804|6.12/44.05/6.26|
> |EGNN|2.7626/4.1565/2.7925|6.10/67.54/5.75|
> |EGNO|2.7609/2.8300/2.7752|6.05/14.26/6.16|
> |FMN|2.7602/2.8430/2.7926|5.46/**8.75**/10.85|
> |EqMotion|2.7598/2.9930/2.7689|5.27/30.40/6.33|
> |ESTAG|2.7603/2.8038/2.7753|5.37/15.55/6.04|
> |FAGNN|2.7597/**2.7946**/2.7875|5.58/10.07/10.99|
> |Ours|**2.7581**/2.7960/**2.7652**|**4.21**/9.46/**4.23**|
>
> *Note: Deterministic baselines are adapted to predict 20 trajectories as in EqMotion for NLL computation. Bond Length indicates chemical bonds (Asp./AT-AT) or skeletal edges (Walk).*
>
> **Q5. Limitation & reproducibility.**
>
> Please refer to our response to Reviewer gjRG (Q2) for a discussion of these concerns and possible future directions. To ensure reproducibility, we will add key implementation details and release our code, datasets, and checkpoints.
>
> **Thanks for the suggestions. We will include these analysis in the revised manuscript.**

---

> > ### Author Rebuttal · Reviewer_FnLA · 2026-04-03
> >
> > Thank you for your response. I would keep my original score.

---

> > > ### Author Response · Authors · 2026-04-04
> > >
> > > Thank you very much for your positive feedback and for your careful consideration of our rebuttal. We will continue refining the paper and ensure that the relevant clarifications and additional results are properly incorporated into the final version. If you have any further questions, we would be very happy to provide additional details.

---

### Official Review · Reviewer_gjRG · 2026-03-18

**Soundness:** 3
**Presentation:** 3
**Significance:** 3
**Originality:** 3
**Overall Recommendation:** 4
**Confidence:** 2

**Summary:**

1. Predicting 3D geometric trajectories demands modeling complex spatiotemporal dynamics while respecting physical symmetries. However, most existing equivariant methods focus on static geometric structures, leaving a critical research gap between static symmetry preservation and the modeling of time-dependent, dynamical SE(3)-equivariant evolution.

2. The authors propose GSE-Flow, an SE(3)-equivariant flow matching framework. It uses Coherent Sequence Encoding and Time-Modulated Embedding to integrate historical and dynamic sequences, using equivariant affine transformations to incorporate velocity and time information, along with a Geometry-Feature Tensorization module that enables Context-Flow Fusion for trajectory evolution.

3. GSE-Flow theoretically ensures SE(3)-equivariance and achieves state-of-the-art performance on geometric trajectory prediction tasks across MD17, MD22, and CMU MoCap datasets. It also shows broad applicability by improving the performance of deterministic baseline models.

**Compliance With Llm Reviewing Policy:**

Affirmed.

**Final Justification:**

The rebuttal has addressed my concerns, and I will maintain my original recommendation.

**Key Questions For Authors:**

1. See weaknesses.

2. Could you provide further insights and experimental comparisons regarding spatiotemporal SE(3)-equivariant, multi-agent, and flow-matching approaches in traffic and autonomous driving contexts?

**Limitations:**

The authors have not provided any limitation analysis of the proposed method.
To better demonstrate its generality and generalization, it would be helpful to discuss its performance issues across different domains and scales, as well as potential future improvements.

**Strengths And Weaknesses:**

Strengths:

1. Existing equivariant flow models are limited to static structures and cannot properly model dynamic temporal data. This work addresses this critical gap by proposing a spatiotemporal SE(3)-equivariant flow paradigm for dynamic systems.

2. To fill this research gap, the authors propose GSE-Flow, an SE(3)-equivariant flow matching framework with three key modules: Coherent Sequence Encoding, Time-Modulated Embedding, and Context-Flow Fusion, to tackle these challenges respectively. Experiments on small molecules, macromolecules, and human motion demonstrate the superior performance of the proposed framework.

3. The authors provide ablation studies to validate the effectiveness of each component for the overall performance.

4. The authors further apply the proposed method as an encoder for various mainstream models to verify its generalizability across different domains.

5. The authors provide solid derivations to prove the theoretical correctness of the proposed SE(3)-equivariant method.

Weaknesses:

1. The authors have not provided failure case analysis to explain sub-optimal performance on certain benchmarks, nor further summarized the limitations of the proposed method for potential future improvements.

2. Compared with deterministic regression methods, this paper proposes a flow-matching method to map from observations to future states, but does not provide any qualitative or quantitative analysis on the multi-modality of the model.

---

> ### Author Rebuttal · Authors · 2026-03-29
>
> **Q1. Failure case analysis on sub-optimal performance.**
>
> We analyze the sub-optimal cases from two perspectives.
>
> 1) For near-deterministic cases with limited uncertainty (e.g., Benzene and Malonaldehyde), deterministic baselines can marginally outperform GSE-Flow, as mapping a Gaussian prior to a concentrated target distribution via ODE integration may introduce additional numerical variance.
>
> 2) For cases exhibiting strong temporal variation (e.g., Buckyball-catcher and Basketball), GSE-Flow achieves better ADE but occasionally sub-optimal FDE. We attribute this to numerical error accumulation during inference. While training optimizes  reconstruction over the full flow time, inference relies on Euler integration; in fast-changing regimes, truncation errors compound over time, resulting in sub-optimal FDE.
>
> **Q2. Limitations & potential improvements.**
>
> We will add a discussion focusing on the following aspects.
>
> 1) As a continuous-time generative model, GSE-Flow incurs higher inference latency due to iterative numerical ODE solving. We plan to use flow distillation for efficient sampling.
>
> 2) While this rebuttal adds multi-modal and physically motivated evaluations, the main paper still primarily relies on ADE/FDE. Future work will incorporate richer multi-modal and physics-informed metrics.
>
> 3) Under severe distribution shifts or long-horizon generation, the model may experience geometric distortions and temporal error compounding. Integrating  kinematic priors and closed-loop training are promising directions.
>
> 4) Applying GSE-Flow to constrained environments like autonomous driving remains challenging. Future work will explore map-aware condition and hierarchical generative priors to scale the vector field estimation.
>
> **Q3. Quantitative analysis on multi-modality.**
>
> To evaluate generation diversity, we compare GSE-Flow with competitive baselines using Average Pairwise Distance (APD) alongside best-of-K prediction accuracy ($\text{minADE}_K$ \/ $\text{minFDE}_K$, $\times 10^{-2}$).
>
> |Method|$\text{minADE}_K\downarrow$ (Asp.\/AT-AT\/Walk)|$\text{minFDE}_K\downarrow$ (Asp.\/AT-AT\/Walk)|$\text{APD}\uparrow$ (Asp.\/AT-AT\/Walk)|
> |:---|:---|:---|:---|
> |ESTAG|6.73\/12.75\/28.28|11.39\/21.79\/47.94|4.48\/16.53\/6.11|
> |EGNN|9.18\/18.61\/52.41|13.95\/30.91\/88.38|17.26\/2.13\/11.30|
> |EqMotion|7.85\/18.53\/1699.73|11.95\/28.75\/1678.68|25.14\/100.57\/**4880.04**|
> |FAGNN|9.19\/18.46\/54.74|13.01\/27.93\/66.01|26.15\/218.72\/416.25|
> |FMN|8.83\/19.02\/48.50|12.42\/27.71\/63.05|23.47\/137.34\/398.41|
> |EGHN|11.28\/16.23\/55.16|16.58\/25.59\/78.81|26.42\/127.13\/214.84|
> |EGNO|7.89\/15.75\/108.62|13.24\/26.72\/195.36|24.47\/120.07\/514.97|
> |Ours|**4.57\/8.97\/24.88**|**7.52\/15.25\/41.15**|**57.08\/246.59**\/666.86|
>
> *Note: We take $K=20$ following EqMotion. For deterministic baselines, we adopt the strategy used in EqMotion to predict multiple trajectories. For our generative model, we generate trajectories by repeated  sampling.*
>
> As shown, some deterministic baselines struggle to produce diverse trajectories (indicated by low APD, e.g., ESTAG and EGNN). While some methods occasionally show higher APD (e.g., EqMotion on Walk), this is accompanied by a significant drop in accuracy (high  $\text{minADE}_K$ & $\text{minFDE}_K$), suggesting a weaker diversity–accuracy trade-off. Our GSE-Flow maintains strong APD while also achieving the best $\text{minADE}_K$ & $\text{minFDE}_K$, suggesting that its diversity is not obtained by sacrificing prediction accuracy.
>
>
> **Q4. Insights & comparisons on SE(3)-equivariant, multi-agent, and flow-matching methods in traffic & autonomous driving (AD).**
>
> To the best of our knowledge, there is limited prior work that directly combines strict SE(3)-equivariant flow matching with AD trajectory prediction. This is partly because vehicle behavior is strongly constrained by HD maps, lane geometry, and traffic rules, which break global SE(3) symmetry. Therefore, directly applying strict SE(3)-equivariant priors without specialized map-aware encoders is likely to be structurally sub-optimal.
>
> To evaluate GSE-Flow in a related multi-agent setting with weaker map constraints, we test it on the ETH-UCY pedestrian datasets (Univ and Zara1) against both generative methods (MoFlow, GeoTDM) and strong equivariant baselines. GSE-Flow achieves competitive accuracy, supporting its potential for modeling multi-agent interactions. Extending it to map-constrained autonomous driving scenarios remains an important direction for future work.
>
> |Model|ADE/FDE (Univ)|ADE/FDE (Zara1)|
> |:---|:---:|:---:|
> |FAGNN|0.53\/1.18|0.38\/0.89|
> |FMN|0.53\/1.30|0.41\/0.93|
> |EGNO|0.51\/1.21|0.41\/0.90|
> |ESTAG|0.51\/1.09|0.38\/**0.81**|
> |EqMotion|**0.50**\/1.10|0.39\/0.86|
> |MoFlow|0.59\/1.28|0.42\/0.91|
> |GeoTDM|0.66\/1.39|0.44\/1.01|
> |Ours|**0.50\/1.07**|**0.37\/0.81**|
>
> **Thanks for the suggestions. We will include these analysis in the revised manuscript.**

---

> > ### Author Rebuttal · Reviewer_gjRG · 2026-04-05
> >
> > Thanks for your response.

---

> > > ### Author Response · Authors · 2026-04-07
> > >
> > > Thank you for your positive feedback and for carefully considering our rebuttal. We will continue refining the paper and ensure that the relevant clarifications and additional results are properly incorporated into the final version.

---

### Decision · Program_Chairs · 2026-04-30

**Decision:**

Accept (regular)

**Comment:**

This paper proposes GSE-Flow, an SE(3)-equivariant flow matching framework for 3D geometric trajectory prediction, combining sequence encoding, time-modulated embeddings, and geometry-feature tensorization to model spatiotemporal dynamics while preserving equivariance. Experiments on MD17, MD22, and CMU MoCap show consistent improvements over prior methods.

Reviewers agree the work is technically sound, well-motivated, and empirically strong, with solid benchmark gains. However, several concerns remain: the generative claim is not fully supported, as evaluation focuses mainly on point-wise metrics without analyzing multimodality or diversity; the empirical study could be strengthened with more ablations, failure case analysis, and clearer efficiency reporting; and the novelty is seen as somewhat incremental, building on existing equivariant and flow-matching designs.

Overall, the paper is well-executed with consistent gains, though limited in validating its generative aspects and depth of analysis. Given the consensus (weak accept) and positive assessment, I lean toward accepting the paper.